# Time-lapsed imaging of nanocomposite scaffolds reveals increased bone formation in dynamic compression bioreactors

Gian Nutal Schädli[1,2], Jolanda R. Vetsch [1], Robert P. Baumann[2], Anke M. de Leeuw[1], Esther Wehrle[1], Marina Rubert[1] & Ralph Müller [1✉]

Progress in bone scaffold development relies on cost-intensive and hardly scalable animal studies. In contrast to in vivo, in vitro studies are often conducted in the absence of dynamic compression. Here, we present an in vitro dynamic compression bioreactor approach to monitor bone formation in scaffolds under cyclic loading. A biopolymer was processed into mechanically competent bone scaffolds that incorporate a high-volume content of ultra-sonically treated hydroxyapatite or a mixture with barium titanate nanoparticles. After seeding with human bone marrow stromal cells, time-lapsed imaging of scaffolds in bioreactors revealed increased bone formation in hydroxyapatite scaffolds under cyclic loading. This stimulatory effect was even more pronounced in scaffolds containing a mixture of barium titanate and hydroxyapatite and corroborated by immunohistological staining. Therefore, by combining mechanical loading and time-lapsed imaging, this in vitro bioreactor strategy may potentially accelerate development of engineered bone scaffolds and reduce the use of animals for experimentation.

[1] Institute for Biomechanics, Department of Health Sciences and Technology, ETH Zurich, Zurich, Switzerland. [2] Particle Technology Laboratory, Department of Mechanical and Process Engineering, ETH Zurich, Zurich, Switzerland. ✉email: ram@ethz.ch

Autologous bone grafts remain to date the gold standard for treating large bone defects[1]. However, the limited availability and poor accessibility of these autografts[1], along with donor site morbidity after harvesting, continue to drive research on engineered bone scaffolds. Mechanical loading plays a vital role in bone remodeling, and it was demonstrated that controlled cyclic loading can improve fracture healing in long bones and may even be used as stimulus where healing is impaired[2,3]. Research strategies for the efficacy testing of bone scaffolds for long bone defects[4] have so far relied on biologically accurate and biomechanically relevant animal experiments[5–7] that are unfortunately expensive, time consuming, and hardly scalable. To emulate mechanical loading in vitro, cell-seeded scaffolds have been cultured using either dynamic perfusion[8] or compression bioreactors[9,10], in some cases a combination of both[11,12]. However, experiments that use bioreactors for cyclic loading[9,10] have merely reported increased expression of osteogenesis markers or end-point calcium levels[9–12] as opposed to in vivo experiments, where micro-computed tomography (micro-CT) has been used to prove the efficacy of the tested bone scaffolds for bone regeneration[5,6]. A holistic engineering approach for novel bone scaffold materials is currently missing and required to translate an engineered product into clinics. Specifically, in vitro efficacy testing of mineral formation under cyclic loading and detailed analysis of formed mineral are needed to close the gap between in vitro and in vivo experiments.

In contrast to bone replacement, for bone repair under mechanical load, a scaffold does not require mechanical stiffness and strength as high as dense bone (elastic modulus = 10–30 GPa[13]) since stabilization is established by fixation. However, a stiffness similar to developing bone[14] with 10–200 MPa compressive moduli[15], porosity above 80%[16], and adequate dimensions[5] are desirable to support healing. For hydrogels in the absence of mechanical load, a stiffness above 225 kPa distinctively improved in vitro osteoblast differentiation and mineralization compared to threefold softer gels[17], as well as increased cell motility[18]. Elastic moduli above 1 MPa have been obtained with synthetic polymers[13]. Several natural polymers have been used, for example silk[19], collagen[20], chitosan[21], and alginate[22]—or synthetic ones such as poly(caprolactone)[23], poly(L-lactic acid)[24], and poly(lactic-co-glycolic acid) (PLGA)[25]. The US Food and Drug Administration approved PLGA and it is one of the most often used biodegradable synthetic polymers[13] as it offers the advantage that its resorption time[26] and mechanical strength[27] can be tailored by the lactic to glycolic acid ratio. However, PLGA exhibits low osteoconductivity[26] and decreasing the molecular weight results in weaker mechanical properties and shorter resorption times[27]. As a solution, PLGA has been used in combination with bioactive filler particles (ceramic or glass) that act as bone-mimicking agents[26] and may also improve the polymer's mechanical properties.

Specifically, bioresorbable PLGA nanocomposites containing hydroxyapatite nanoparticles have been extensively used to prepare bone scaffolds[26]. Often, only the bioactive property of hydroxyapatite nanoparticles is used to render PLGA scaffolds osteoconductive, either by hydroxyapatite coating[28] or admixing[25]. To also exploit the reinforcement effect of the nanoparticles, higher particle contents are desirable while at the same time, particle agglomeration within the polymer matrix should be avoided[29]. Avoiding particle agglomeration is more challenging at higher filler contents, requiring lengthier and more energy intense processing to distribute particles within the polymer matrix[29].

This work aims at producing reinforced nanocomposite scaffolds with mechanical properties suitable for cyclic loading by merely modifying the widely used solvent casting particulate leaching (SCPL) method[24]. We hypothesized that in scaffolds cultured in dynamic compression bioreactors under cyclic loading, more mineral is formed by extracellular matrix (ECM) mineralization than under static conditions. The formed mineral is longitudinally monitored and quantified by time-lapsed micro-CT imaging to evaluate mineral maturation[30] and bone volume (BV)[31]. Furthermore, we used the time-lapsed images to calculate a spatial bone formation rate (BFR) that, in combination with immunohistological images, allows us to distinguish cell-mediated mineral from medium precipitation. We showed that this bioreactor approach enabled comparison between scaffolds containing pure hydroxyapatite and a mixture of hydroxyapatite and barium titanate. The piezoelectric barium titanate is attractive for bone repair due to its ability to deliver additional electric stimulation[32]. Altogether, we demonstrate a holistic and rigorous in vitro testing framework for bone scaffold development with efficacy testing of mineral formation that aims to close the gap between in vitro and in vivo experiments by detailed microstructural analysis of in vitro formed mineral.

## Results and discussion

**Nanoparticle reinforced polymer nanocomposite bone scaffolds.** For the nanocomposite bone scaffold, we chose commercial hydroxyapatite nanoparticles[25] with specific surface area (SSA) of 55 m$^2$/g (HA55) and with 20 m$^2$/g (HA20). Both powders exhibited the typical X-ray diffraction (XRD) pattern for hydroxyapatite (Fig. S1a). The HA20 nanofillers were spherical (Fig. S1a, left inset), while HA55 were needle-like (Fig. S1a right insets). The particles are polycrystalline as the HA55 had crystal size $d_{XRD} = 24$ nm, well below their visible length (Fig. S1a, right insets, 100 nm in average), and the $d_{XRD} = 49$ nm of HA20 was well below the average particle diameter ($d_{BET} = 100$ nm), calculated from nitrogen adsorption.

Using these nanoparticles, we prepared bone scaffolds with a composition of 9:1:1 wt. ratio of NaCl porogen, polymer, and hydroxyapatite filler, resulting in nanocomposites with 30 vol% filler loading[25] and with 6 mm diameter and 12 mm height, envisioned for critical-sized long bone defects. We modified the standard SCPL[25] by ultrasonication to improve the dispersion of nanofillers in the PLGA matrix[33] and employed pressure molding to ensure interconnectivity between the porogens[34]; such scaffolds are denoted with a "(u)". The porosity of the scaffold was analyzed by high-resolution micro-CT (Fig. 1a). The generated 3D model of a scaffold section[31] (Fig. 1b) showed that after NaCl leaching, the pressure-molded scaffold was highly porous. The calculated porosity of about 83% was consistent with previous reports using this composition[25]. The ultrasonication was successful to deagglomerate the nanofillers[35] as they were well distributed and exposed at the surface (Fig. 1b, zoom-in, arrows), which is preferential to enhance the bioactivity of a PLGA scaffold[28]. Due to the shape of the NaCl porogen, the nanocomposite exhibited cube-like pores with sizes from 240 to 440 µm (Fig. S1c), similar to the size of the sieved NaCl (250–315 µm).

The dependence of the scaffolds' mechanical properties on the SCPL preparation method was investigated with HA55 nanoparticles. Figure 1c shows compressive stress as a function of strain of dry and unseeded nanocomposite scaffolds with 2:1 and <1:1 height to diameter aspect ratios. Scaffolds made with the standard SCPL method having a 2:1 aspect ratio (Fig. 1c, green broken line, n = 2) showed no identifiable linear elastic region. Thus, no compressive modulus could be calculated. Figure 1a–I shows one of these scaffolds after compression. The top half of the scaffold collapsed during compression, while the lower half seemed intact, becoming mushroom-like. This was most likely caused by non-uniform porogen distribution during casting[36],

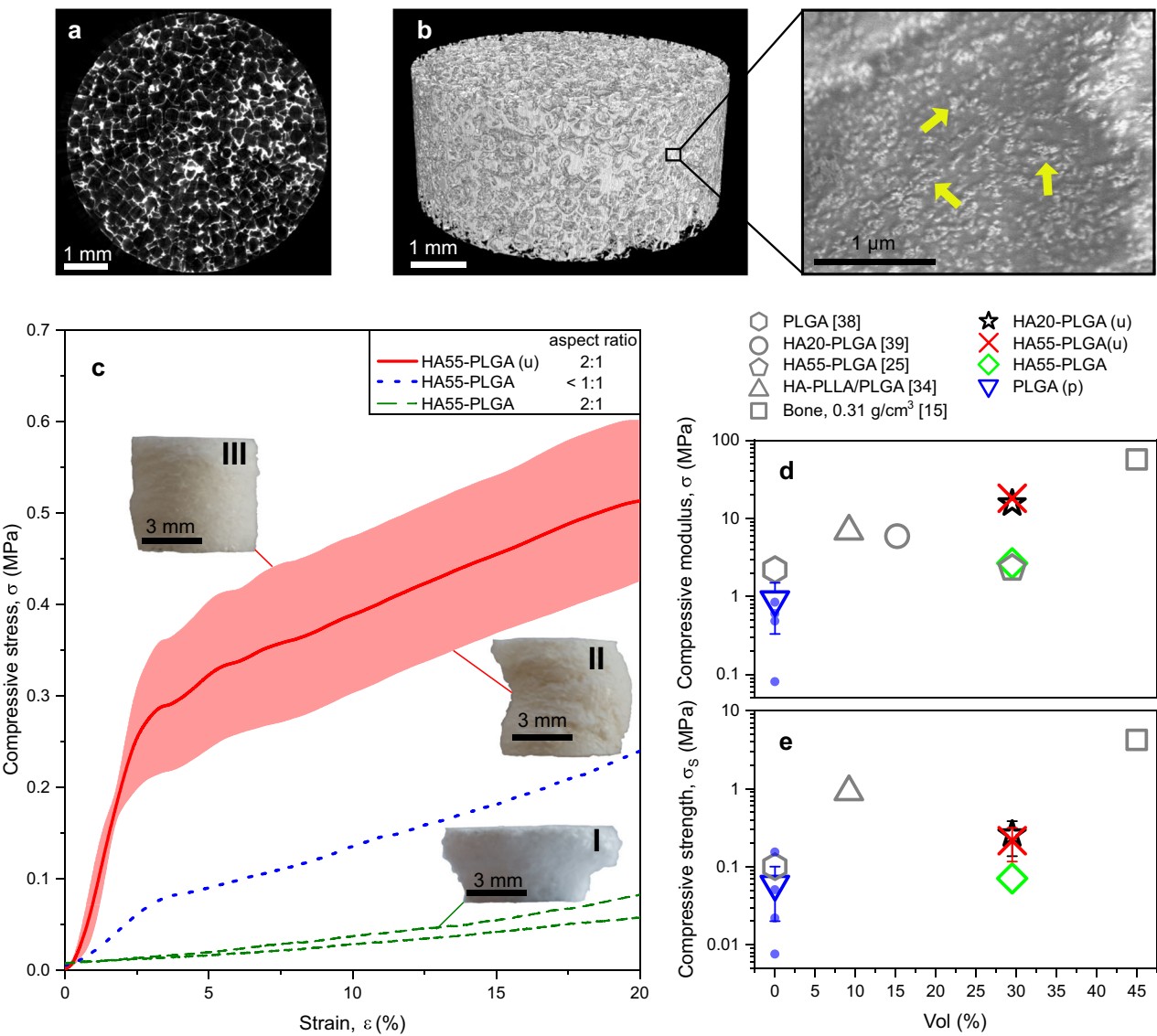

**Fig. 1 Reinforced polymer nanocomposite bone scaffold properties. a** Micro-CT raw image of a cross-sectional slice of a representative HA55-PLGA (u) scaffold. **b** 3D model obtained from that micro-CT scan, along with a scanning electron microscopy image of the surface, showing hydroxyapatite nanoparticles exposed at the surface (arrows). **c** The compressive stress as a function of strain of dry and unseeded HA55-PLGA (u) nanocomposite scaffolds with a 2:1 height to diameter aspect ratio (solid red line, $n = 5$), and scaffolds made by standard SCPL[25] with aspect ratio <1:1 (blue dotted line, $n = 1$) or 2:1 (green broken lines, $n = 2$). The red line represents the average and the shaded areas the s.d. The letter u denotes the modified production method with ultrasonication and pressure molding. Because of the high aspect ratio some scaffolds buckle (**a-II**) vs. (**a-III**), resulting in lower compressive strength and variation after 2.5% strain. **d** Compressive moduli and **e** strength of bone[15] or dry and unseeded nanocomposite scaffolds of various filler specific surface areas as a function of filler vol% and similar ones from literature[25,34,38,39]; all with porosity above 80%. The letter p denotes production by pressure molding. Symbols and error bars of tested samples represent the mean ± s.d.; triangle down $n = 4$, star $n = 5$, cross $n = 5$, diamond $n = 1$ independent samples. Error bars within the symbols are not shown.

limiting standard SCPL to scaffolds with low aspect ratios[37]. The scaffold made with an aspect ratio <1:1 (Fig. 1c, blue dotted line, $n = 1$) had a compressive modulus of 2.7 MPa, consistent with the literature[25]. However, scaffolds with such properties using the standard SCPL were not reproducible using our high aspect ratio molds, as reported already[37].

Scaffolds prepared by the modified SCPL (Fig. 1c red or Fig. S2 black solid line, each $n = 5$) had a 2:1 aspect ratio and exhibited reproducible performance. A small variation was observed at ca. 2.5% strain due to the high aspect ratio, which resulted sometimes in buckling (Fig. 1c-II vs. III). The scaffolds with smaller SSA particles (HA20-PLGA (u)) showed a similar stress–strain curve (Fig. S2) to HA55-PLGA (u). These scaffolds showed high compressive moduli of 18.16 ± 2.35 MPa (Fig. 1c, red solid line,

$n = 5$) or 15.62 ± 2.89 MPa (Fig. S2, $n = 5$). We compared the mechanical properties of these dry and unseeded scaffolds (Fig. 1d: cross, star) with published PLGA-based dry bone scaffolds that have porosities larger than 80% to shed light on the influence of filler vol%, SSA, and processing. The reference scaffolds were prepared by gas-foaming particulate leaching[25] (pentagon), high-pressure compression molding[34] (triangle-up), SCPL[38] (hexagon), or 3D[39] bioplotted (circle). Without filler (triangle down), the mechanical properties were not improved by pressure molding. These pure PLGA scaffolds (triangle down) produced by the modified SCPL had a compressive modulus similar to the literature[38] (hexagon). The small difference is accounted to the higher molecular weight PLGA employed there[38]. The impact of the modified SCPL on the compressive

**Table 1 Prepared polymer nanocomposite scaffolds.**

| Sample name | SSA, m²/g | Shape | $d_{BET}$, nm | $d_{XRD}$, nm | $d_{z\text{-}aver}$ nm |
|---|---|---|---|---|---|
| HA20-PLGA (u) | 19.19 ± 0.36 | Spherical | 100.59 ± 1.89 | 49 | 320 ± 13 |
| HA55-PLGA | 54.74 ± 0.78 | Needle-like | 35.25 ± 0.51 | 24 | 303 ± 13 |
| HA55-PLGA (u) | // | // | // | 24 | |
| B3H7 | 15.62 ± 0.06 | Spherical | 63.81 ± 0.23 | 70 | 147.6 ± 3.2 |

The spherical hydroxyapatite particles have a smaller specific surface area (SSA) and correspondingly larger equivalent particle size ($d_{BET}$) and crystal size ($d_{XRD}$) than the needle-like nanoparticles. Ultrasonication and pressure-molded samples are denoted with (u). The average hydrodynamic agglomerate size $d_{z\text{-}ave}$ were comparable for HA.

modulus for the H55-PLGA (cross vs. diamond) was discussed above. A slightly smaller compressive modulus is shown by HA20-PLGA (u, star) that could be associated with the smaller SSA[33]. The reported HA-PLLA/PLGA (triangle-up[34]) had lower compressive modulus, despite being prepared by a high-pressure melt molding procedure. This inferior property is likely due to the much larger fillers[33] with 6.5 μm agglomerates[34] compared to the sub-micron sized agglomerates used in this study (Table 1), lower filler vol% and lack of proper dispersion of nanoparticles in the polymer matrix. Low filler vol% and lack of proper dispersion also resulted in lower compressive modulus for the reported HA20-PLGA (circle)[39] composite; noteworthy, they used hydroxyapatite (supplier and product number) identical to our HA20-PLGA (u, star) composite.

These results indicate that ultrasonication of high SSA (20, 55 m²/g) nanofillers and SCPL modified by pressure molding enables synthesis of PLGA nanocomposites scaffolds with up to 30 vol% hydroxyapatite filler fraction, porosities above 80% and large aspect ratios (Fig. 1c-III). Using this approach we prepared reinforced polymer nanocomposite bone scaffolds that exhibited compressive moduli (cross, star) that were, at least, twofold higher than in previous reports (pentagon[25], circle[39], triangle-up[34]), reducing the difference in properties between such biodegradable scaffolds and cancellous bone (square). Also, the compressive strength of our nanocomposite scaffolds (cross, star) was three times higher than when made by standard SCPL (diamond). However, the compressive strengths (Fig. 1e) were still much lower than cancellous bone (square), but the PLLA/PLGA polymer mixture (triangle-up) indicates that this property could be improved by changing the host polymer. The dry mechanical tests limit the comparison to in vivo conditions, for which the scaffolds should be typically soaked for 24 h prior to quasi-static compression testing.

**Monitoring scaffolds in dynamic compression bioreactors.** For the bioreactor culture, scaffolds were seeded with human marrow stromal cells (hMSCs). Each bioreactor contained two scaffolds (Fig. 2a) and was mounted into a self-made mechanical stimulation unit (MSU)[40]. The MSU allowed controlling the force $F_{thres}$ to contact the scaffold, as well as frequency and strain (Fig. 2a) of the loading regime. One scaffold was cultured under static conditions and the other was loaded cyclically three times per week for 5 min. First needle-like HA55-PLGA (u) nanocomposite scaffolds have been chosen because the employed nanoparticles are established in literature[25]. Before loading, the scaffolds were contacted using $F_{thres} = 0.05$ N, so that during the full displacement (Fig. 2c, ca. 180 μm) the specified target strain (Fig. 2c, 5%) was reached, resulting in a peak of the force measurement (Fig. 2d). Therefore, during half of the loading cycle, the piston was not in contact with the scaffold (Fig. 2d, 0 N force). Using $F_{thres} = 0.05$ N, scaffolds were loaded first with a loading scenario of 1 Hz and 5% strain and then in a second independent experiment with 5 Hz and 3% strain. These loading strains are at the edge (3%) or outside (5%) of the linear elastic regime of the

scaffolds (Fig. 1c). The influence of cyclic compression to changes in scaffold height and max. force was further investigated in different scaffold compositions. We compared mixtures (3:7 vol. ratio) of barium titanate and hydroxyapatite scaffolds (B3H7) to HA20-PLGA (u) scaffolds, denoted as HA scaffolds. The barium titanate and hydroxyapatite nanoparticles were of comparable size and spherical shape to exclude any potential size or shape effects. Before the compression bioreactor culture, all produced scaffolds were first tested dry using non-destructive stress–strain measurements. The analysis of the stress–strain measurement using $F_{thres} = 0.05$ N (Fig. S3a) showed that the scaffold was contacted in the toe region (Fig. S3a). Therefore, we chose a higher $F_{thres} = 0.2$ N to measure the compressive modulus (Fig. S3b). For the compression bioreactor culture, scaffolds were selected to have an equal compressive modulus across all groups (Fig. S3c).

Typical for polymer foams[41], the force response decayed rapidly during the first few compression cycles (Fig. 2c) and then in a second phase decreased very slowly (Fig. 2d, representative sample), showing that the mechanical integrity of the scaffold was maintained during the culture. On the other hand, each scaffold (individual symbols in Fig. 2e, f) typically lost height over the course of the cell culture. This height loss was a bit more pronounced (not significant) with the loading scenario 1 Hz, 5% strain (Fig. 2e) compared to the 5 Hz, 3% strain (Fig. 2f, g) and is attributed to the regular cyclic loading. The height reduction of up to 17% (Fig. 2e, circle) after 7 weeks of cyclic loading (three times per week of 5 min long) was still smaller than for cell-seeded electrospun calcium phosphate-PLGA nanocomposite scaffolds[12], which showed a 30% height reduction after 9 days of daily cyclically loading for 10 min with 1 Hz and 5% strain. On the last day of the experiment, B3H7 scaffolds cultured under 5 Hz, 3% strain showed a significantly lower ($p = 0.011$) height reduction than HA scaffolds cultured under 1 Hz, 5% strain. Figure 2f shows the change of the max. force response over the course of the culture. For scaffolds contacted with $F_{thres} = 0.05$ N (Fig. 2h, i), the max. force typically decreased with time until after 3–4 weeks it remained relatively stable for each scaffold (individual symbols). In contrast to the scaffold height, scaffolds loaded with 5 Hz, 3% strain (Fig. 2i) exhibited lower max. force values. A $F_{thres} = 0.2$ N contact force resulted in relatively stable max. force values throughout the experiment, while it did not affect the change in scaffold height. Therefore, variations of samples observed in Fig. 2f, h (triangle, pentagon) are attributed to a $F_{thres}$ that is not adequately adjusted to the scaffold's mechanical properties. In addition, such variations will be increased by scaffold surfaces parallel to the piston not being perfectly plane and aligned.

**Cell distribution, time-lapsed micro-CT imaging, and longitudinal monitoring.** The DNA amount was quantified after 1 day and at the end of the culture for comparison with previous reports[10,12,19,42] that cultured human cells in bone scaffolds. Both scaffold groups cultured under static and dynamic conditions

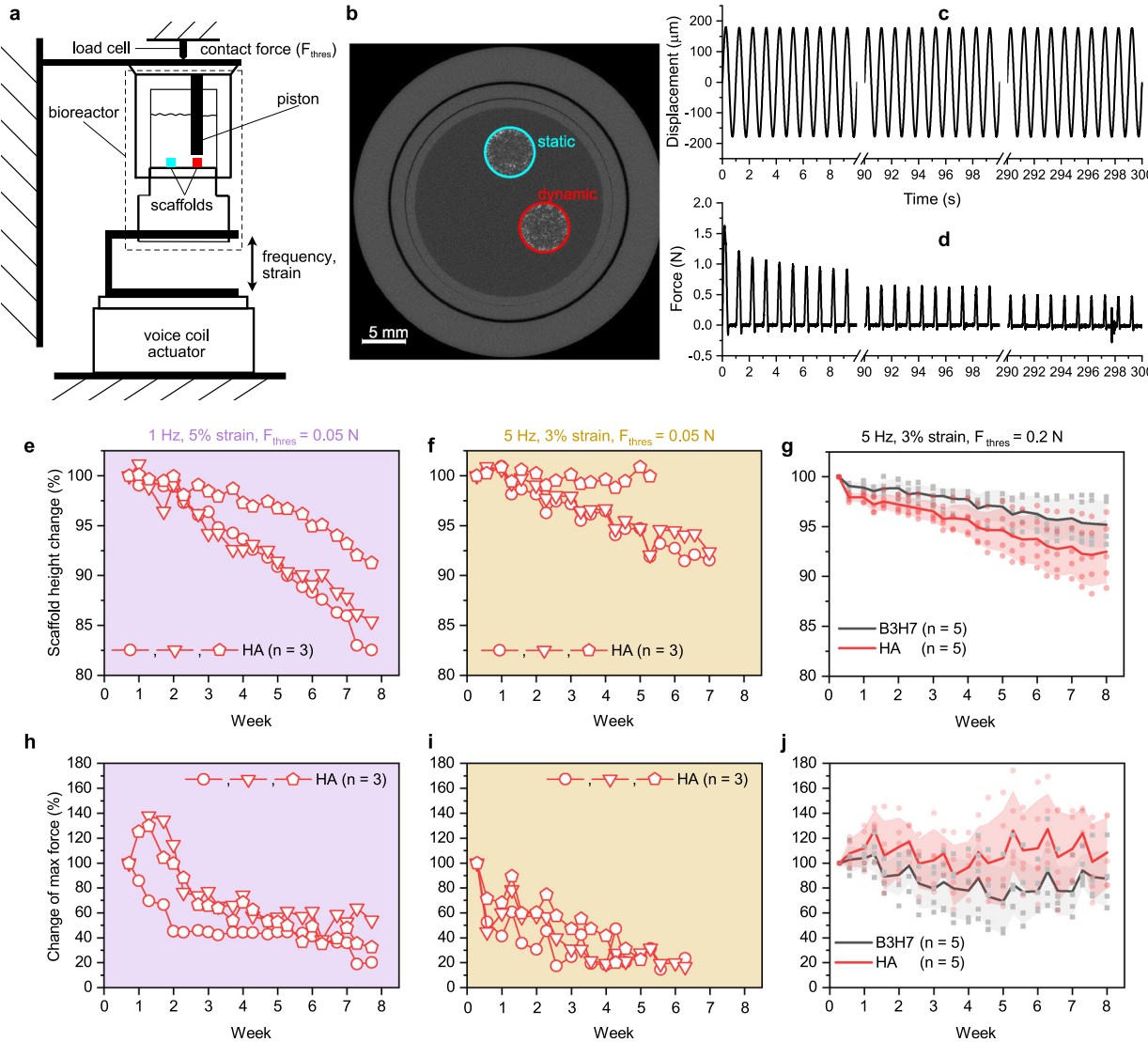

**Fig. 2 Monitoring of cell-seeded scaffolds in dynamic compression bioreactors. a** Scheme of two scaffolds fixed inside the bioreactor[40]. The red scaffold is loaded cyclically. **b** Raw micro-CT image of the bioreactors showing the two scaffolds. **c** Displacement recording during cyclic compression with 1 Hz, and 5% strain and **d** the corresponding force recording. Height change of scaffolds containing pure hydroxyapatite (HA) and a mixture of barium titanate and hydroxyapatite (B3H7) during dynamic culture with 1 Hz, and 5% strain (**e**), 5 Hz, 3% strain (**f**), both with $F_{thres} = 0.05$ N, the latter also with $F_{thres} = 0.2$ N (**g**). **h–j** The corresponding change of max. force response. Symbols represent data points, bold lines the mean and shaded areas the s.d. No significant difference was found between B3H7 and HA scaffolds on the last day; $p > 0.05$; t-test.

(1 Hz, 5% strain), showed a reduction in DNA by a factor of 3.4 and 4.8, respectively (Fig. 3a). The static group retained significantly ($p = 0.027$) more cells after 54 days of culture (Fig. 3a). A DNA reduction during culture was also observed for SCPL-made PLGA scaffolds[42], silk scaffolds[19], electrospun PLGA with calcium phosphate nanoparticles[12], and hydrogels cultured under cyclic loading[10]. The DNA reduction observed in this study and literature[10,12,19,42] occurred for scaffolds cultured with (hMSCs) or adipose-derived stem cells under perfusion or static conditions, independent of cell culture medium used. Thus, there seems to be no general trend in DNA change with respect to culture conditions and cell culture medium. The seeding density used in this study and literature[10,12,19,42] was rather high, with more than one million cells per scaffold. We believe that especially the hydrophobic nature of PLGA scaffolds[43] resulted in weak cell attachment to the scaffold and therefore caused a drop in DNA. Under dynamic culture conditions, there was a significant smaller DNA amount compared to static condition. This additional decrease

may have resulted from enhanced mineralization as a fraction of the osteoblasts undergo apoptosis after completing their bone-forming function[44].

Figure 3b shows a hematoxylin and eosin (H&E) stained cross-section of a HA55-PLGA (u) scaffold after 54 days of static culture and Fig. 3c after dynamic culture. The cells successfully penetrated scaffold pores but remained mainly in the top half, regardless of the culture condition. Non homogeneous cell distributions using static cell seeding by pipetting were also reported previously[45]. We attribute the cell distribution to the small 30 μl droplet seeding volume that is about one-third of the scaffold's pore volume. However, seeding of larger droplets was difficult because the scaffold would not readily absorb the droplet due to its hydrophobic nature[43]. Histology sections stained with H&E showed production of a dense ECM (magnification of Fig. 3b, c) within the scaffold's pores.

Using time-lapsed micro-CT scans, mineral maturation and mineral formation were analyzed. Figure 3d–g shows

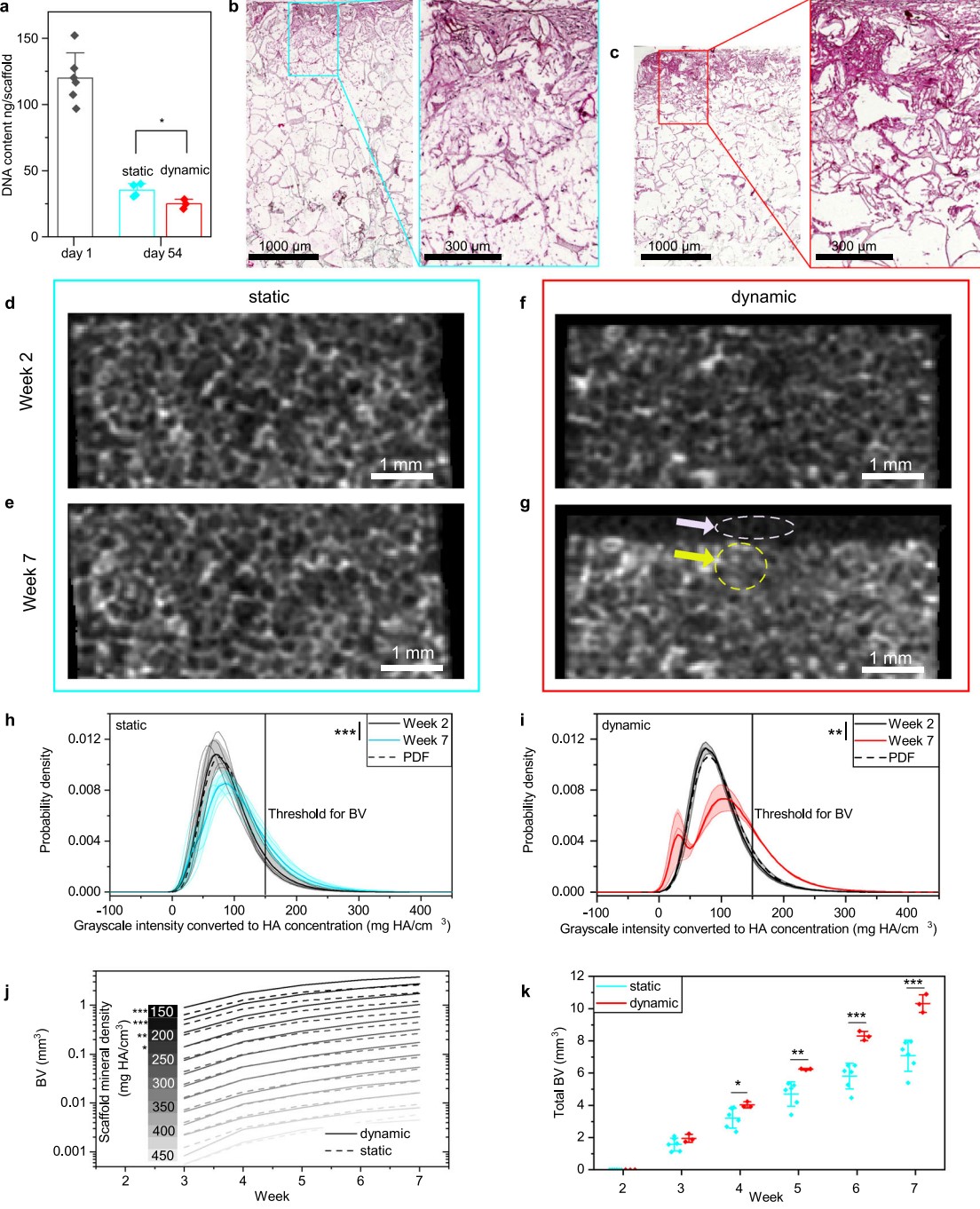

**Fig. 3 Cell distribution, time-lapsed micro-CT monitoring of HA55-PLGA (u) scaffolds, 1Hz and 5% strain. a** DNA content per scaffold at day 1 ($n = 6$) and 54 (static $n = 4$, dynamic $n = 3$). Columns represent mean and error bars the s.d.; *$p < 0.05$; $t$-test. **b** Microscopy image of a hematoxylin and eosin stained vertical cross-section in a scaffold cultured under static condition and **c** under dynamic condition. **d** Representative micro-CT cross-sectional slice of a scaffold cultured under static condition from week 2 and **e** week 7, respectively, of a scaffold cultured under dynamic condition (**f, g**). **h** Histogram of 3D grayscale images of static ($n = 6$) and dynamic (**i**, $n = 3$) cultured scaffolds. Lines represent the mean and the shaded area the s.d. Dashed line represents the probability density function (PDF) for week 2; **$p < 0.01$, ***$p < 0.001$; paired $t$-test. **j** Average bone volume (BV) as function of culture time for mineral densities above the threshold shown in **h**; *$p < 0.05$ $t$-test for week 7, dynamic vs. static. **k** Total BV as function of time for static ($n = 6$) and dynamic ($n = 3$) culture conditions. Symbols represent the mean and error bars the s.d. *$p < 0.05$, **$p < 0.01$, ***$p < 0.001$; $t$-tests were used to highlight differences between both conditions for each time-point. Shapiro–Wilk normality and Levene's test for equality of variances were not significant.

representative noise filtered cross-sectional images from such scans of HA55-PLGA (u) scaffolds after 2 weeks (Fig. 3d, f) and 7 weeks (Fig. 3e, g) of static (Fig. 3d, e) and dynamic (Fig. 3f, g) cell culture. The intensities of the voxels in the grayscale images correspond to the material density in mg HA/cm³. In contrast to previous studies on silk scaffolds[45], the bone nanocomposite used

here is visible under micro-CT (Fig. 3d, f) due to the hydroxyapatite nanofiller, interfering with established scaffold image processing methods[46]. Thus, a refined procedure based on the histogram of time-lapsed micro-CT images is required to distinguish bone nanocomposite material from newly formed mineral. Figure 3h, i shows histograms as normalized density

distributions of the grayscale images from weeks 2 to 7 (Fig. 3d–g).

Images from week 2 were chosen as reference because at that time-point the scaffolds were completely soaked with medium. Mineral formation is differentiated from scaffold by shift in peak position and increasing width of the histogram between time-points as maturation of mineralizing ECM progresses, increasing grayscale intensity.

First, we analyzed the density distributions of our reference images from week 2 by fitting a three-parameter lognormal probability density function[47] (PDF, dashed line). Under static conditions (Fig. 3h, dashed line), the lognormal PDF has median $m = 118.56 \pm 14.76$ mg HA/cm$^3$, geometric standard deviation $\sigma_g = 1.40 \pm 0.04$, shift parameter $\theta = -29.94 \pm 6.74$ mg HA/cm$^3$ and under dynamic conditions (Fig. 3i, dashed line) $m = 122.67 \pm 11.86$ mg HA/cm$^3$, $\sigma_g = 1.38 \pm 0.05$, $\theta = -30.63 \pm 8.95$ mg HA/cm$^3$. Thus, after 2 weeks there was no observable difference between scaffolds cultured under static and dynamic conditions.

Second, we compared the density distributions from week 2 (Fig. 3h, i black line) to week 7 (Fig. 3h, i red line) to track formed mineral. This comparison also reveals the height loss of those scaffolds (Fig. 3i) compacted during dynamic culture conditions (1 Hz, 5% strain) because the applied strain was close to or outside of the linear elastic region (Fig. 1c). Scaffolds cultured under static conditions maintained their geometry (Fig. 3d vs. e). The height loss of compacted scaffolds (Fig. 3f vs. g) was accounted for by keeping the analyzed volume of interest, the so-called total volume (TV), constant for subsequent images of a scaffold. After 7 weeks, the probability density increased for densities above 100 mg HA/cm$^3$ for both culture conditions (Fig. 3h, i) which corresponds to whitish pixels (yellow arrow) in Fig. 3d–g. This effect was more pronounced under cyclic loading. As a result of scaffold compaction, the probability density increased under dynamic conditions for mineral densities from 0 to 40 mg HA/cm$^3$ (Fig. 3i), representing the larger number of blackish pixels (Fig. 3g, white arrow). Mere scaffold compression results in a negligible widening of the density distribution (Supplementary Note 1, Fig. S4). The probability for voxels exhibiting a mineral density >150 mg HA/cm$^3$ was significantly larger at week 7 compared to week 2 for both static ($p = 0.005$) and dynamic ($p = 1E-5$) conditions. Based on the density distribution, we chose a global threshold of 150 mg HA/cm$^3$, which is higher than in previous reports (97.5[8], 130[46] mg HA/cm$^3$), allowing us to obtain the BV. These thresholds were chosen to distinguish mineralized ECM from the background, e.g., culture medium, and corresponded to small mineral nodules[8]. Here, a slightly higher threshold was chosen to reduce partial volume effects because the scaffolds were already pre-mineralized due to the embedded nanoparticles. Figure 3j shows the BV as function of time for scaffold mineral density (SMD, bins with width of 25 mg HA/cm$^3$) from 150 to 450 mg HA/cm$^3$. For scaffolds cultured under dynamic conditions, starting from week 3, more mineral matured to a density of 225 mg HA/cm$^3$ compared to static condition. At later time-points, mineral matured to 350 mg HA/cm$^3$. The differences in BV between the dynamic and static condition were statistically significant ($p = 0.0004, 0.0006, 0.009, 0.047$) at week 7 for all mineralization levels up to 250 mg HA/cm$^3$, where most of the mineral formation (BV > 0.1 mm$^3$) also occurred. This in vitro formed mineral density is consistent with mineral formation in silk scaffolds[46], however, it is still well below mineralized tissue formation during non-critical-sized in vivo defect healing[30]. In a mouse defect model, mineral matures from around 400 to 700 mg HA/cm$^3$ during healing[30]. Figure 3k shows the total BV as function of time of the scaffolds cultured under static or dynamic conditions with 1 Hz and 5% strain. Longitudinal monitoring showed after 4 weeks a significant ($p = 0.049$) difference in BV that steadily increased

with culture time. Thus, it is crucial to capture the maturation process of formed mineral to understand underlying mechanisms[30]. The stimulatory effect of cyclic loading on mineral formation has so far only been reported for scaffolds tested in vivo[48]. The results presented in Fig. 3 were reproduced for scaffolds cultured with a 5 Hz and 3% strain loading scenario (Supplementary Note 2, Fig. S5).

**Micro-CT monitoring, image registration, and local mineral formation analysis.** Next, we analyzed the mineral formation in more detail to calculate the BFR and bone resorption rate (BRR) in scaffolds cultured under static or dynamic (5 Hz, 3% strain) conditions. Figure 4a shows cross-sectional images that were registered to week 2, with mineral formation (orange), quiescent (gray), and mineral resorption (blue) sites. The registered images reveal that under dynamic conditions more mineral was formed than under static conditions. In vivo, the blue sites correspond to bone resorption[49], however, here are no cells that could resorb formed mineral or scaffold material. Therefore, we associate the blue sites to scaffold degradation, deformation, or registration error. For example, the images of scaffolds cultured under dynamic conditions show some sites (center, upper half of image) that are labeled as quiescent material in week 3, and then in week 4 as removed mineral. Consequently, any changes in the quiescent (gray) sites may also be considered as an error because they should stay constant. Also, because scaffolds under dynamic conditions are compacted during culture, we applied a manual non-rigid registration[50] before automated registration that was accomplished by stretching the scaffolds in later time-points to the height of week 2 assuming a linear deformation. This correction was needed to avoid mistakenly labeling voxels as mineral formation or resorption sites because of scaffold deformation.

We have spatially quantified the mineral formation (Fig. 4b, open symbols) and resorption (Fig. 4b, filled symbols) volumes for each week by normalizing the volumes with a volume of interest, corresponding to the top or bottom half of the scaffold (Fig. 4a). This analysis showed that mineral formation within the scaffolds was strictly linear (adj. $R^2 = 0.998 \pm 0.003$, $n = 14$) consistent to in vivo fracture healing in mice[30] where low density bone was formed linearly from weeks 2 to 4 until mineral maturation started. There was no difference between static and dynamic samples regarding linearity, independent of the analyzed region (top or bottom). On the other hand, as expected, there was no clear trend for mineral resorption (adj. $R^2 = 0.698 \pm 0.43$, $n = 14$). We then calculated the BFR and BRR from the slopes to compare between static and dynamic conditions as well as between the top and bottom half of the scaffold. Under static condition, there was no difference in BFR between the top and bottom half. Under dynamic condition, however, the BFR in the top half was 1.75-fold higher than under static condition ($p = 0.013, 0.016$) and also 1.3-fold higher than in the bottom half (no significant difference). In contrast, no difference in BRR, which was slightly negative, was found between the different groups. Note that BRR values were about a tenth of BFR values. Also, the coefficient of variation (CV, Table S1) of the quiescent volume within the selected volume of interest was for all samples ≤2.4%— smaller than the CV = [10.7%, 30.4%] of the reported BFR values. Therefore, any erroneous contribution from scaffold shrinkage, deformation, or registration error toward BFR was within the standard deviation of the measurement and is thus negligibly small. As this was not the case for scaffolds loaded with 1 Hz and 5% strain, they were not considered for image registration. As the H&E stained histology sections (Fig. 3b, c) showed that most of the hMSCs were located in the top half of the scaffold, we conclude that the higher BFR under cyclic loading originates from

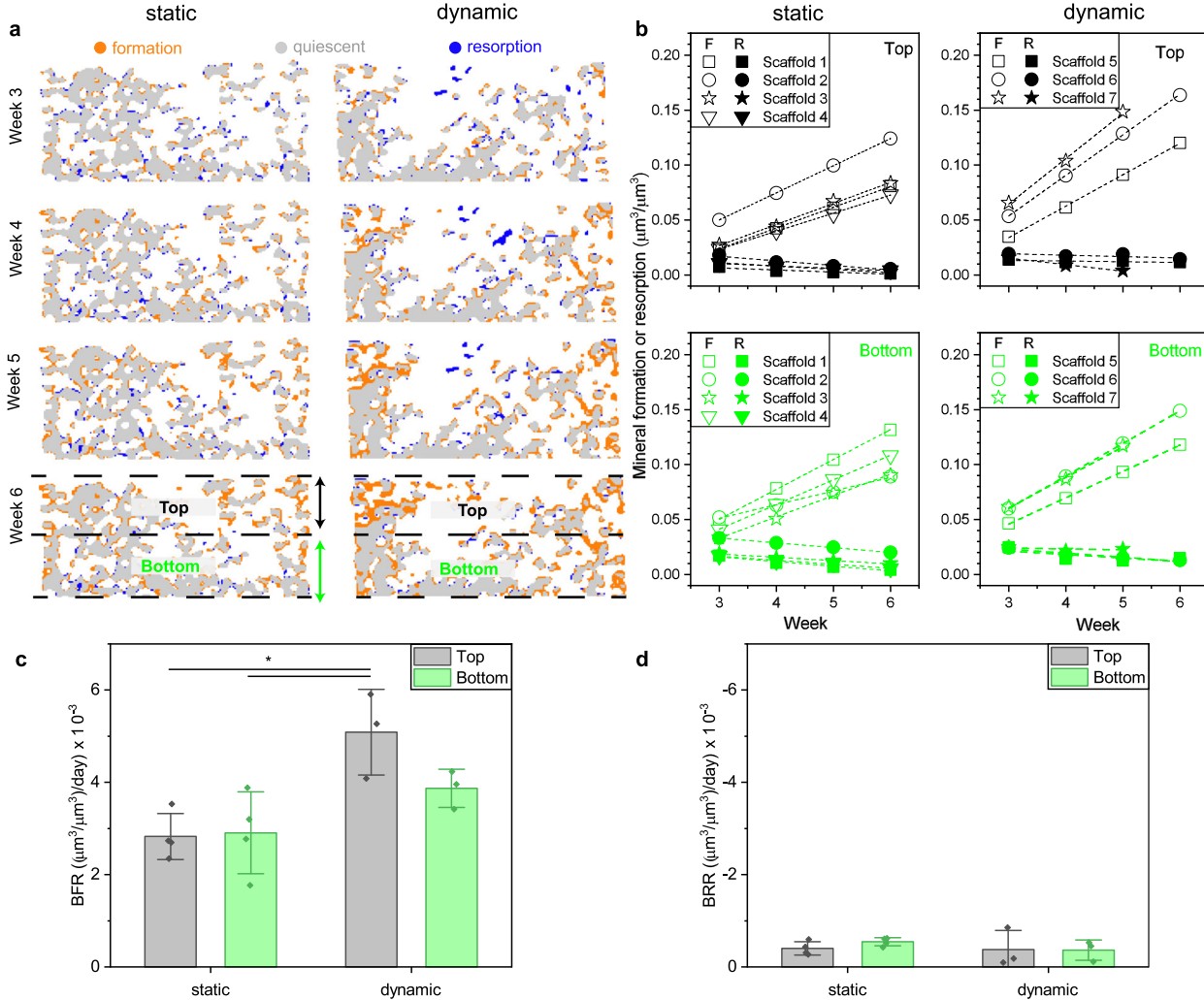

**Fig. 4 Micro-CT monitoring, image registration, and local mineral formation analysis of HA55-PLGA (u) scaffolds. a** Color-coded vertical cross-sectional images for static and dynamic (5 Hz, 3% strain) cultured scaffolds after 3, 4, 5, and 6 culture weeks. The orange, blue, and gray colors represent mineral formation, resorption, and quiescent volumes, respectively. **b** Absolute mineral formation (F, open symbols) and resorption (R, filled symbols) volumes normalized with the analyzed total volume of 3D images (**a**) as function of time. The total volume was split in a top (black) and bottom (green) region (**a**). **c** Bone formation rate (BFR) per day obtained from linear fits (adj. $R^2 = 0.998 \pm 0.003$, $n = 14$) of the slopes (open symbols) shown in **b**; static ($n = 4$), dynamic ($n = 3$). **d** Bone resorption rate (BRR) per day obtained from linear fits (adj. $R^2 = 0.698 \pm 0.43$, $n = 14$) of the slopes (filled symbols) shown in **b**; static ($n = 4$), dynamic ($n = 3$). *$p < 0.05$; one-way analysis of variance (ANOVA) and Bonferroni post hoc tests. Shapiro–Wilk normality and Levene's test for equality of variances were not significant. Data are shown as mean ± s.d.

enhanced cell-mediated mineral maturation. Therefore, time-lapsed micro-CT monitoring and image registration[49] followed by local mineral formation analysis may also be used in combination with histology to distinguish between mineral formed from ECM mineralization or medium precipitation[51].

**Comparison of hydroxyapatite and barium titanate/hydroxyapatite scaffold materials**. We next evaluated if time-lapsed micro-CT also allows to compare different bone scaffolds, e.g., scaffolds containing a barium titanate and hydroxyapatite mixture. The piezoelectric property of barium titanate makes it attractive for bone repair due to its ability to deliver additional electric stimulation under cyclic loading. The high linear absorption properties of barium titanate (Fig. S6) challenges assessment and comparison to widely used hydroxyapatite-based scaffolds by end-point micro-CT. The scaffolds were first cultured for 4 weeks in control medium. Afterwards, the cells were cultured in osteogenic medium until week 8. For the analysis of the micro-CT scans, a volume of interest in the top 1 mm region of

the scaffold was chosen because during the culture in control medium air bubbles were observed mainly in the bottom part of the scaffold (Fig. S7a–d). As the histogram also revealed (Fig. S7k, l) the presence of air bubbles in the top 1 mm region in the scan from week 1, only scans after week 2 were considered for the analysis. The same global threshold of 150 mg HA/cm$^3$ was applied for both materials. Figure 5a shows the total BV for B3H7 and HA scaffolds. During the culture in control medium, only HA scaffolds exhibited an increase in the total BV that became more distinct in osteogenic medium. Analysis of the BV growth rate (Fig. 5b), which is similar to the BFR (Fig. 4c) due to negligible BRR (Fig. 4d), shows that HA scaffolds had already a significantly ($p = 1E-5$, $2E-5$, $2E-5$, $4E-5$) higher rate than B3H7 scaffolds during the culture in control medium. Regardless of the scaffold material composition, however, the BV growth rates of scaffolds under dynamic conditions were almost identical to those cultured under static condition. Switching the culture medium to osteogenic medium had the highest effect on BV growth rate for HA scaffolds under dynamic conditions. For

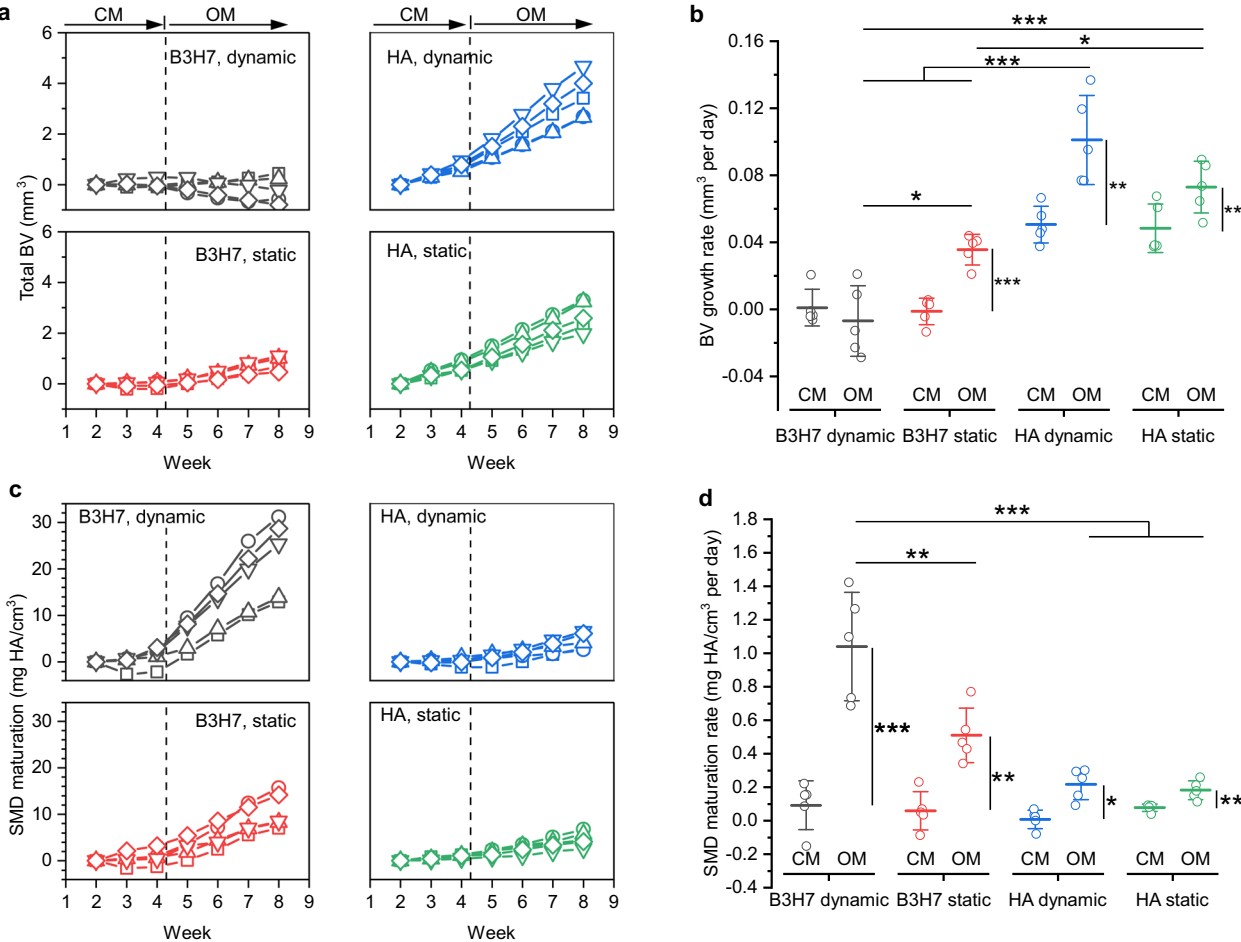

**Fig. 5 Mineral formation kinetics in B3H7 and HA scaffolds. a** Total bone volume (BV) for scaffolds containing pure hydroxyapatite (HA) and a mixture of barium titanate and hydroxyapatite (B3H7) under dynamic and static conditions ($n = 5$ for each group). On day 30 (vertical dashed line), the culture medium was switched from control medium (CM) to osteogenic medium (OM). **b** BV growth rates during culture in CM and OM obtained from linear fits of the data points in **a**. **c** Scaffold mineral density (SMD) maturation. **d** the corresponding rates. *$p < 0.05$, **$p < 0.01$, ***$p < 0.001$; longitudinal groups were tested with paired $t$-test, one-way analysis of variance (ANOVA) with post hoc Bonferroni correction to compare multiple groups separated by CM or OM. In **b** for CM, the groups HA dynamic/static are significantly different from B3H7 dynamic/static; ***$p < 0.001$. Shapiro–Wilk normality test was not significant. Data are shown as mean ± s.d.

B3H7 scaffolds, switching to osteogenic medium had only an effect for scaffolds cultured under static conditions.

Time-lapsed micro-CT imaging allowed to track individual cell-seeded scaffolds and observe their response to changing culture conditions. Previously, it was shown that micro-CT data ($R^2 = 0.96$) correlates with calcium assay levels[51]. Cell-seeded silk scaffolds cultured in control medium inhibited spontaneous mineralization[51]. In this work, HA scaffolds exhibited a measurable increase in BV during culture in control medium but there was no observable difference between dynamic and static culture conditions. Therefore, the height loss (Fig. 2e) due to the cyclic loading had no measurable contribution to the scaffolds, corroborating the fact that the increased BV (Fig. 3k) and BFR (Fig. 4c) are due to cell-mediated mineralization. When the culture medium was switched to being osteogenic, this increase was reproduced for HA scaffolds (Fig. 5b) but this increase was not significant, which can be attributed to the comparatively shorter time in osteogenic medium. When the medium was switched to being osteogenic, the BV growth rate only increased for B3H7 scaffolds cultured under static condition. Under dynamic conditions, the BV growth rate showed a small decrease (Fig. 5b), which was attributed to the height loss (Fig. 2e). A comparison between HA and

B3H7 scaffolds based on BV is somewhat limited because B3H7 scaffolds exhibited right from the beginning more voxels above the threshold (Fig. S7i, j vs. k, l) due to the high absorbing barium titanate (Fig. S6). Thus, any mineralization that could occur inside a voxel would not be captured by the BV analysis if that voxel was already above the threshold. Therefore, we analyzed next the SMD, conserving the density information of a voxel. During the culture in control medium, all scaffolds exhibited barely any change in SMD (Fig. 5c, regardless of culture condition). Once the scaffolds were cultured in osteogenic medium, B3H7 scaffolds exhibited a strong SMD maturation (Fig. 5c). The SMD maturation rate increased longitudinally, which was significant ($p = 0.001$, 0.004, 0.02, 0.006) for all scaffolds and culture conditions (Fig. 5d). However, B3H7 scaffolds showed the strongest effect on media change. The SMD maturation rate under static condition was more than 2x (not significant) and under dynamic more than 5x ($p = 2E-5$) higher than for HA scaffolds under dynamic condition. After the culture, we examined those scaffolds by histology. Picro Sirius Red staining showed a distinctively denser collagenous ECM under dynamic conditions compared to static condition, both in B3H7 and HA scaffolds (Fig. 6a, c vs. b, d). Interestingly, birefringence of collagen fibers, which indicates thicker collagen fibrils[52] was

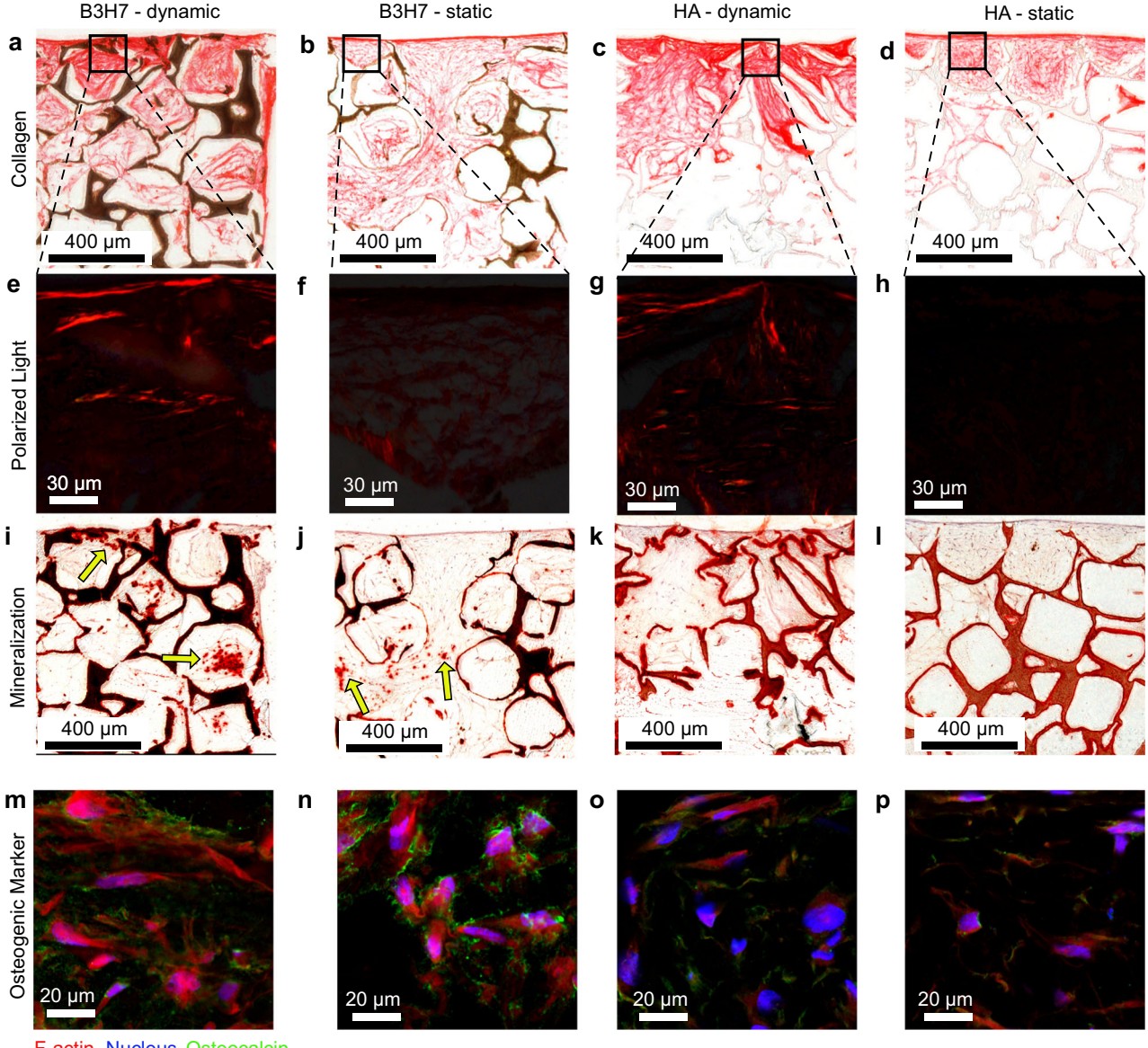

**Fig. 6 Extracellular matrix (ECM) products in response to cyclic loading. a–d** collagen in red as stained by Picro Sirius Red in B3H7 scaffolds under dynamic, static, and HA scaffolds under dynamic as well as static conditions. The scaffold structures were stained brownish to black. **e–h** Birefringent collagen fibrils of marked regions in **a–d**. **i–l** Mineralized ECM stained by Alizarin Red. Arrows (**i**, **j**) point to mineral clusters formed in pores. **m–p** Immunostaining images of F-actin, nucleus, and osteocalcin.

only observed when the scaffolds were cultured under cyclic loading conditions (Fig. 6e, g, vs. f, h). The smaller DNA content in loaded scaffolds (Fig. 3a) and collagen staining results suggests that cyclic loading stimulated collagen secretion by osteoblasts but did not result in increased cell number. We then confirmed mineralization by Alizarin Red S staining (Fig. 6i–l) in subsequent sections. The mineralized ECM was stained (red) in all scaffolds and co-localized with the collagen staining, confirming cell-mediated mineralization. In accordance with the SMD matura-tion analysis (Fig. 5d), in the B3H7 scaffolds some pores (arrows, Fig. 6i, j) showed intensely stained mineral clusters that were larger in scaffolds cultured under dynamic (Fig. 6i) than static (Fig. 6j) conditions. In addition, B3H7 scaffolds enhanced formation of mineral clusters as compared to HA scaffolds either under static or dynamic conditions. Immunohistochemistry confirmed the presence of osteocalcin, a biochemical marker for osteoblasts in all scaffolds (Fig. 6m–p). In line with the SMD

maturation analysis and mineral nodule formation observed from Alizarin Red S staining, the amount of osteocalcin appears greater in B3H7 scaffolds than HA scaffolds, irrespective of loading condition.

For more than 20 years, research efforts on bone regeneration have been made[24]. However, despite the endeavor to support healing of broken bones by synthetic materials, no scaffold for critical-sized long bone defects has made it into clinics. This work presents a complete path from scaffold production, followed by in vitro testing that combines time-lapsed micro-CT imaging with longitudinal monitoring of cell-seeded bone scaffolds within a relatively short time. Our results suggest that the mechanical[14] and design requirements[16] mentioned earlier are met by PLGA nanocomposites that were prepared by the modified SCPL method and contain 30 vol% high SSA nanofillers. We tested these mechanically competent nanocomposite bone scaffolds and tracked their efficacy for mineral formation under cyclic loading,

which has so far only been available to animal studies. In this in vitro study, the density of formed mineralized tissue in HA scaffolds did not reach in vivo values. On the other hand, in critical-sized in vivo bone defects, polymer nanocomposite scaffolds containing tricalcium phosphate[5] or hydroxyapatite[6] particles were without bone morphogenetic proteins also not able to restore the defect with higher density bone (>500 mg HA/cm$^3$). Thus, we believe the formed mineralized tissue of lower density is due to the insufficient osteoinductive properties of HA scaffolds. The B3H7 scaffolds exhibited a significantly ($p = 0.003$, $2E − 5$, $1E − 5$) enhanced SMD maturation rate (Fig. 6d), which is likely due to the formation of mineral clusters in pores. Time-lapsed micro-CT imaging of scaffolds in dynamic compression bioreactors is therefore an effective strategy to identify promising candidates for bone tissue engineering applications. Our approach bridges the gap between the latest in vitro[10] and in vivo[6] research and is also able to facilitate the understanding of the underlying mechanism behind bone formation and its dependence on material interaction. We have shown that BFR and SMD were significantly higher in scaffolds cultured under dynamic conditions compared to static culture. This stimulatory effect of cyclic loading also resulted in enhanced collagen secretion, corroborating in vitro mineral formation by hMSCs. In the future, this functional in vitro testing framework could accelerate the development of novel engineered bone biomaterials or scaffolds that interact with pharmaceutical therapies.

## Methods

**Nanoparticle characterization.** The hydroxyapatite nanoparticles were purchased from Berkeley Advanced Biomaterials Inc. (BABI-HAP-N100) and Sigma-Aldrich (677418). Their crystal size ($d_{XRD}$) was determined by XRD (XRD, Bruker AXS D8 Advanced or D2 Phaser) and Topas 4 software by fitting hexagonal hydroxyapatite (ICSD 082289). The SSA was measured by five-point $N_2$ adsorption at 77 K (Micromeritics Tristar II Plus) after degassing for at least 1 h at 150 °C. The hydrodynamic agglomerate size was measured by dynamic light scattering using a Zetasizer (Nano ZS, Malvern Instruments) right after deagglomeration by ultrasonication (Sonics). The barium titanate nanoparticles were from Sigma-Aldrich (467634) and elsewhere in detail characterized[53].

**Polymer nanocomposite scaffold fabrication.** Scaffolds were prepared by either standard SCPL method[25] or modified by additional ultrasonication (Sonics) for deagglomeration and employing pressure during molding to establish interconnection between the NaCl porogen (S7653, Sigma-Aldrich) that were sieved from 250 to 315 μm. According to SCPL[25], PLGA (Resomer® RG 756 s, 76,000–115,000 g/mol, Sigma-Aldrich) was dissolved in dichloromethane (CH$_2$Cl$_2$, 270997, Sigma-Aldrich) to obtain a 10% (w/v) solution and then the nanofiller was added. This solution was mixed at 1000 rpm for 15 min (LLG-uni*THERMIX* 1) and cast into cylindrical Teflon molds of 6 mm diameter and 12 mm height containing NaCl porogens and stirred with a spatula until the mixture thickened, resulting in nanocomposites with a 1:1:9 wt. ratio between PLGA, nanofiller, and NaCl. These scaffolds were dried for 24 h at ambient conditions and then for 48 h under $10^{−3}$ mbar vacuum.

For the modified SCPL method, hydroxyapatite (10% w/v) in CH$_2$Cl$_2$ was ultrasonicated for 90 s with an appropriate tip probe (pulse 3 s on, 2 s off) in an ice bath. The PLGA was added afterwards to avoid its degradation[54]. Smaller volumes up to 2 ml were mixed overnight at 1000 rpm (LLG-uni*THERMIX* 1). Then, the mixture was cast into the Teflon molds containing the porogens, stirred with a spatula until the mixture thickened, and molded with about 0.3 MPa. The pressure during molding was measured with a hydraulic load cell (tecsis, F1119/P3962). Larger batches (5–10 ml) were produced by mixing ultrasonicated nanoparticles with PLGA in a dual asymetric centrifuge[55] (FlackTek SpeedMixer 150 FVZ) at 3000 rpm until the polymer dissolved. After adding the porogens, the composite was mixed again at 3000 rpm until thickening, thereafter, the mixture was molded similarly to before.

The pressure-molded nanocomposites were dried first in the mold overnight at ambient conditions, then the scaffolds were removed from the mold and dried for another night at ambient conditions, before drying for at least 24 h under vacuum (<0.2 mbar). Scaffolds of 6 mm diameter were cut into cylinders of 12 mm height for stress–strain measurement and 4 or 3 mm height for cell cultivation with 1 Hz, 5% strain or 5 Hz, 3% strain loading, respectively. The porogen was leached by placing the scaffolds in Milli-Q® (18.2 MΩ.cm, Merck Millipore) water for about 30 h. During porogen leaching, the water was regularly replaced. The volume fraction of the nanofiller embedded within the polymer was calculated using densities of $\varphi_{HA} = 3.1$ g/cm$^3$ and $\varphi_{PLGA} = 1.3$ g/cm$^3$, respectively.

The process was further upscaled to produce long scaffolds in ferritic nitrocarburized steel molds of 40 mm height, 6 mm inner diameter (X153CrMoV12, Thyssenkrupp). Chloroform (C2432, Sigma-Aldrich) was used to dissolve PLGA of higher inherent viscosity PLGA (Resomer® LG 855 S, Evonik). Consistent to before, the composite was mixed until a homogeneous and viscous mixture was obtained. These mixtures contained less solvent before molding, the porogen content was increased to result in hydroxyapatite nanocomposites with a 1:1:12 wt. ratio between PLGA, nanofiller, and NaCl. Thus, the mass loss after porogen leaching was held constant at ca. 88.6 ± 1.4%, consistent to before. The solid filler volume fraction in scaffolds containing barium titanate and hydroxyapatite was held constant at ca. 30 vol%.

Table 1 lists all nanocomposite scaffolds employed here, labeled with the embedded nanofiller (HA) and SSA in m$^2$/g. For example, HA55-PLGA denotes nanocomposites made with hydroxyapatite having 55 m$^2$/g SSA. Nanocomposites produced with the modified SCPL method are labeled with ultrasonication and pressure molding (u). For nanocomposites containing a 3:7 vol. ratio of barium titanate and hydroxyapatite (B3H7), the barium titanate nanoparticle properties are given.

**Scaffold porosity.** Selected nanocomposite scaffolds were scanned at high-resolution (5 μm) on a μCT 50 (Scanco Medical). The energy was set to 55 kVp and intensity to 200 μA. An integration time of 1000 ms and a frame averaging of 5 was used. Images were subsequently processed with Fiji (ImageJ 1.51p) and BoneJ (BoneJ[56] 1.4.2) and the porosity (ε) was calculated based on the BV to TV ratio as $ε = 1 − BV/TV$. The nanoparticles and their distribution in the polymer matrix were imaged by scanning electron microscopy (Hitachi S-4800, 3 kV). Pore sizes were determined using Fiji (ImageJ 1.51p) by measuring 80 pores.

**Mechanical properties of reinforced polymer nanocomposite scaffolds.** Compressive stress–strain curves of the nanocomposites were obtained at dry state immediately after their creation following the ASTM F2150/D695 standard using a specimen size with a 2:1 height to diameter ratio. Briefly, the nanocomposites were compressed unconfined at a crosshead speed[19] of 0.5 mm/min using an Instron MicroTester equipped with a 500 N load cell, resulting in compressive stress–strain curves typical for elastic–plastic foam material[57]. The compressive Young's Modulus was determined from the slope of the linear elastic region and the compressive strength at the intersection of the tangents of the linear elastic and the collapse regions[57]. Non-destructive stress–strain measurements were conducted with a Zwick Z006 equipped with a 10 N cell. The scaffolds were compressed at a crosshead speed of 0.3 mm/min and preload of 0.2 N. Raw measurement data were processed with MATLAB R2019a.

**In vitro cell culture.** Before cell seeding, the scaffolds were attached to polysulfone substrates using polydimethylsiloxane (10:1 ratio, Sylgard® 184 Elastomer Kit, VWR) and sterilized with H$_2$O$_2$ plasma[58] at 50 °C. The cell line was obtained from fresh human bone marrow (Lonza Walkersville, Inc) and the following surface antigens were confirmed: CD14+, CD31−, CD34−, CD44+, CD71+, and CD105+[45]. The hMSCs were tested negative by a PCR mycoplasma testing kit (ATCC 30–1012 K) as well as DNA Hoechst staining. Passage 3 hMSCs[45] were recovered and expanded[19]. Briefly, the cells were expanded under standard culturing conditions (37 °C, 5% CO$_2$) for 7 days in expansion medium composed of Dulbecco's Modified Eagle Medium with high glucose and pyruvate (DMEM, 41966, Gibco, Thermo Fisher Scientific), 10% fetal bovine serum (FBS, 10270106 Gibco, Thermo Fisher Scientific), 1% antibiotic-antimycotic (Anti-Anti, 15240062, Gibco, Thermo Fisher Scientific), 1% non-essential amino acids (11140035, Gibco, Thermo Fisher Scientific) and 1 ng/ml basic fibroblastic growth factor (PHG0369, Gibco, Thermo Fisher Scientific). After this period, the cells were counted and resuspended at a concentration of 2.8 or 1.6 million cells per 30 μl in control medium (DMEM, 10% FBS, 1% Anti-Anti). Scaffolds were seeded by pipetting 30 μl cell suspension on top of each scaffold and incubation in a six-well plate for 90 min to allow for cell adhesion. Afterwards, 8 ml control or osteogenic medium, composed of control medium with 50 μg/ml L-Ascorbic acid-2-phosphate sesqui-magnesium salt hydrate (A8960, Sigma-Aldrich), 100 nM dexamethasone (D9184, Sigma-Aldrich) and 10 mM β-glycerophosphate (410990250, Acros Organics, Thermo Fisher Scientific) was added. The next day the scaffolds were transferred into the in-house designed bioreactors[40] where they, depending on the experiment, were cultured in control or osteogenic medium. During the cell experiment, the medium was exchanged three times per week.

**Surface functionalization.** Scaffolds that received surface functionalization were immersed in an aqueous 1 wt% Poly(vinyl alcohol) solution (22225, MW 6000, 80% hydrolyzed, Polysciences Inc.) for 15 min at the end of the salt leaching process to render the surface hydrophilic[59]. After H$_2$O$_2$ plasma sterilization, scaffolds were surface functionalized[60] by pipetting 40 μl of 1 mg/ml Arginylgly-cylaspartic acid peptide motif (4008998, GRGDS, Bachem) and 2 mg/ml dopamine hydrochloride (H8502, Sigma) in 10 mM, pH 8.5 Tris buffer (T6791, Sigma-Aldrich). The scaffolds were incubated for 2 h at 37 °C and then washed 3x with PBS and then stored in an incubator (37 °C, 5% CO$_2$) until cell seeding. The surface functionalization was conducted on HA and B3H7 scaffolds.

**Table 2 Cell culture conditions during bioreactor culture.**

| Bioreactor culture conditions | Scaffold | Number of seeded cells (Mio) | DNA, day 1 (ng/scaffold) |
|---|---|---|---|
| 1 Hz, 5 % strain ($n = 4$) | HA55-PLGA (u) | 2.8 | – |
| Static, no loading ($n = 6$) | HA55-PLGA (u) | 2.8 | 120.08 ± 17.24 ($n = 6$) |
| 5 Hz, 3 % strain ($n = 3$) | HA55-PLGA (u) | 1.6 | – |
| Static, no loading ($n = 4$) | HA55-PLGA (u) | 1.6 | 54.96 ± 9.81 ($n = 4$) |

Each bioreactor housed two scaffolds, where one scaffold was loaded with cyclic compression.

**Time-lapsed bioreactor and imaging**. The bioreactors allowed to perform cyclic loading and time-lapsed micro-CT monitoring of the scaffolds[40]. A custom-made MSU controlled by LabVIEW National Instruments was used to apply displacement-controlled load on the bioreactor. Each bioreactor housed two scaffolds. One of the two scaffolds was loaded with a sinusoidal compression cycle at 1 Hz and 5% strain (1st experiment), or 5 Hz and 3% strain (2nd experiment) for 5 min three times a week, while the other scaffold served as static control. The culture conditions are summarized in Table 2. The height of the loaded scaffolds was measured before every loading sequence by contacting them with 0.05 N ± 0.01 N. In a 3rd experiment, 5 Hz and 3% strain with 0.2 N contact force were applied. During cyclic loading, the force was continually recorded. The scaffolds were monitored using a µCT40 (Scanco Medical) once a week[46]. The bioreactors were scanned at 45 kVp energy and 177 µA intensity. The integration time was set to 300 ms and a frame averaging of 2 was applied. Because of a technical issue with an aging X-Ray tube, for the 2nd experiment the bioreactors had to be scanned at 165 µA. The total energy was kept constant by using 215 ms integration time and a frame averaging of 3. The resulting image resolution was 36 µm. Using these scan settings, the bioreactors were out of the incubator for about 40 min. Scaffolds cultured under dynamic conditions that were compressed >5% strain due to malfunction of the MSU were excluded ($n = 2$).

**Cell proliferation assay**. The cell proliferation was quantified by Quant-iT™ PicoGreen® dsDNA Assay Kit at day 1 and the last day of the experiment. The scaffolds were rinsed with (PBS), after which the cells were lysed by adding 1.5 ml 0.2% (v/v) Triton X-100 (X100, Sigma-Aldrich) solution in aqueous 5 mM MgCl₂ (208337, Sigma-Aldrich) solution. Then, the scaffolds were disintegrated three times for 10 s at 25,000 rpm in a Mini-Beadbeater-1 (Biospec) to expose the dsDNA. After incubation at room temperature for 48 h, the samples were centrifuged at $3000 \times g$, 5 °C (Hettich Mikro 200 R) for 10 min. The DNA assay was performed according to the manufacturer's instructions. The fluorescence was measured with a plate reader (Tecan Spark 10 M) at an excitation wavelength of 480 nm and a detection wavelength of 520 nm. The DNA amount per scaffold was determined with a standard curve.

**Histology and immunohistochemistry**. The scaffolds were fixed in 10% (v/v) neutral buffered formalin for 4 h and embedded (PrestoChill, Milestone) for cryo-sectioning. Vertical cross-sections were cut approximately through the middle of the scaffold at a thickness of 5 µm. Cell nuclei and ECM were stained with hematoxylin (Mayer's, Sigma-Aldrich) and eosin (Y disodium salt, Sigma-Aldrich). Mineral and collagen were visualized on 10 µm thick vertical, sequential cross-sections. Mineral was stained by 2 mg/ml Alizarin Red S (A5533, Sigma-Aldrich) for 1 min. Collagen was stained by 1 mg/ml Picro Sirius Red (365548, P6744, Sigma-Aldrich) for 1 h. Histology sections were imaged with a Slide Scanner Pannoramic 250 (3D Histech) at ×20. Polarized light microscopy images were taken in transmission mode with a Zeiss AxioImage.Z2 running ZEN Blue (Zeiss) at ×40 0.75 NA. For immunohistochemistry, all washing and staining was performed at room temperature. Primary antibodies were diluted in PBS containing 1% bovine serum albumin (BSA, Sigma-Aldrich). Prior to staining, cryosections were permeabilized with 0.1% Triton X-100 in PBS for 10 min and blocked with 3% BSA for 1 h. Immunostaining of osteocalcin was performed by incubating with anti-osteocalcin (1:200; Abcam ab93876) overnight. After washing in PBS three times, samples were incubated with secondary antibody Alexa Fluor-647 IgG H&L (1:1000; Abcam ab150075) for 1 h. F-actin was stained with Phalloidin (1:50; Sigma-Aldrich P1951) for 1 h. After washing in PBS three times, cell nuclei were stained with Hoechst 33342 (1:200; Sigma-Aldrich B2261) for 10 min. Immuno-histochemistry staining was validated by a secondary antibody control without adding the primary antibody and a negative control where no antibody was added. Immunohistochemistry sections were imaged with a Leica TCS SP8 setup running Leica LAS AF SP8 software version 4.0. Images were taken at ×63 1.4 NA. All images were processed with Fiji.

**Image processing, registration, and bone formation rate**. Voxels in grayscale images were converted to corresponding hydroxyapatite densities (mg HA/cm³) based on calibration measurements using a phantom[61]. All images were processed with a constrained Gaussian filter[46] (sigma 1.2, support 1) using IPL Scanco AG software V5.42. Histograms were generated using uniform bins of width 25 mg

HA/cm³. For the TV the scaffold was centered within a cylindrical mask with 6 mm diameter. A global threshold of 150 mg HA/cm³ was used to determine BV. Unfiltered images were used to superimpose images of one scaffold from different time-points, so-called registration, using an intensity-based least-squares algo-rithm[49]. The image from week 2 served as base for registration of follow-up scans. Images from follow-up scans of scaffolds under cyclic loading were stretched before registration, using a linear interpolation (Python 3.6.6, Scipy 1.1.0) to match the scaffold's height to week 2. Registered images were Gaussian filtered and a global threshold was applied. The registration process provided a three-colored image containing voxels that are present only in week 2, only in follow-up weeks or in both time-points; representing mineral resorption, formation or quiescent volumes. In contrast to bone remodeling[49], the BFR was defined as the amount of mineral formation per time normalized with the analyzed TV and the respective BRR as the amount of mineral resorption per time normalized with the analyzed TV. Changes in the quiescent volumes were considered as registration error and quantified as the coefficient of variation, which is the standard deviation of the quiescent volumes from the different time-points normalized by the average value of the quiescent volumes.

**Statistics and reproducibility**. All data are represented as mean ± standard deviation. Unpaired two-tailed Student's $t$ tests were performed to compare two groups. One-way ANOVA with subsequent Bonferroni post hoc testing was performed for more than two groups. The normality was tested with Shapiro–Wilk and where indicated homogeneity of variances with Levene's test. Longitudinal comparison was done with paired Student's $t$ tests. Differences were considered statistically significant when $p < 0.05$. The exact number of independent samples ($n$) is indicated in the figure legend. All statistic tests were performed with Rstudio 1.1.456 or OriginPro 9.5. Normalized histograms of week 2 were fitted (Python 3.6.6, Scipy 1.2.0) by a three-parameter lognormal distribution[47]:

$$f(x) = \frac{e^{-\ln\left(\frac{x-\theta}{m}\right)^2 / (2\sigma^2)}}{(x - \theta)\sigma\sqrt{2\pi}} \tag{1}$$

where $\theta$ denotes the shift parameter, $m$ the median, and $\sigma$ the standard deviation of the log of the distribution, therefore the geometric standard deviation $\sigma_g = \exp(\sigma)$.

**Reporting summary**. Further information on research design is available in the Nature Research Reporting Summary linked to this article.

## Data availability

All the data underlying the graphs are included in Supplementary Data 1. Other information that support the findings of this study are available from the corresponding author upon reasonable request.

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

## Acknowledgements

The research was funded by ETH Zurich (grant no. ETH-17 15-1). We truly appreciate Sotiris E. Pratsinis (Laboratory of Particle Technology, ETHZ) for his critical comments on nanocomposite and project design and his support during the initiation phase. We thank the Optical Materials Engineering Laboratory (ETHZ) for providing SEM instrumentation and Christoph Blattmann (ETHZ) for taking the images. We acknowledge the support of the Scientific Center for Optical and Electron Microscopy ScopeM (ETHZ) for providing confocal microscopy instrumentation. Thanks also go to Martin Ehrbar (University Hospital Zurich, USZ) for providing access to sterilizing facilities, Krishnar Mahendraraja (Central Sterilization Supply Department of USZ) for sterilizing the scaffolds and to Julia Griesbach as well as Bryant Schroeder (all ETHZ) for stimulatory discussions. A python framework to process micro-CT images was made available by Nicholas Ohs (ETHZ).

## Author contributions

G.N.S., J.R.V., and E.W. conceived the project. R.P.B. and G.N.S. produced the scaffolds and performed the mechanical characterization. G.N.S., R.P.B., and J.R.V. performed the in vitro cell experiments. G.N.S. performed and analyzed with R.M. the micro-CT images. G.N.S., J.R.V., R.P.B. wrote the initial draft of the manuscript. A.M.D.L. performed histological and immunohistochemical staining as well as confocal microscopy. M.R. supervised histology and immunohistochemistry experiments. R.M. supervised the project until completion. All authors discussed the results, commented on the manuscript, and contributed to its final version.

## Competing interests

The authors declare no competing interests.
