## [Peer Review File · Communications Biology]

Reviewers' comments:

Reviewer #1 (Remarks to the Author):

This is well designed and performed project. Time-lapsed microCT imaging of cultured scaffolds is innovative. There are no major concerns about the methods and results. But, it would be more interesting to show if the dynamic loading enhanced osteoblast number in the scaffold compared to the static culture.

Reviewer #2 (Remarks to the Author):

The manuscript "Time-lapsed microstructural imaging of polymer nanocomposite scaffolds in dynamic compression bioreactors reveals increased bone formation and mineralization under cyclic loading" by Muller et al. present a bioreactor system that combines mechanical loading with longitudinal microCT imaging to assess bone mineralization in a PLGA scaffold reinforced with hydroxyapatite nanoparticles. This novel approach allows for the rapid and rigorous evaluation of engineered bone scaffolds performance in vitro.

Although the use of real time microCT imaging for the evaluation of engineered bone grafts is an exciting approach, the results shown in this manuscript are too preliminary. The authors claim that their approach can identify good candidates for bone replacement, but this is not shown in the manuscript so far. The authors study only one type of scaffold, and the increased bone formation due to cyclic compression measured by microCT is not validated with any other techniques that demonstrate that this correlates with better performance. Addressing some of the scaffold weaknesses pointed out by the authors (low compressive strengths, low mineral density after culture, weak cell attachment, uneven cell distribution) and/or comparing the performance of different scaffolds in their bioreactor system would demonstrate the potential of this approach and strengthen the relevance of this manuscript.

1. Supplementary Fig 1 and 2 were not provided to the reviewers.
2. Authors mention how traditional in vitro assays evaluate bone engineered scaffolds based on markers of osteogenesis and end-point calcium levels. It would be interesting to see those in this paper to be able to compare with previous studies and see how the osteogenesis markers and calcium levels correlate with the increased bone density measured in this study. Similarly, further histological analysis of the samples to show new bone formation and colocalization with cell clusters will strengthen the results.
3. The authors explain that the mineral density formed in these scaffolds under cyclic loading is still well below mineralized tissue formation during in vivo defect healing. What's the authors hypothesis to explain this? What do these differences between in vivo and in vitro behavior mean in terms of the value of this bioreactor testing strategy?

Minor comments

1. Based on the methods, I understand that the scaffolds analyzed in figure 2 were seeded with bone marrow stromal cells, but not for the analysis performed in Fig 1. This should be made clear in the text.
2. It is not clear what the Fig 1b zoom in is trying to show. Adding some arrows or clarifications might help
3. In Page 14 Line 1, where it says Fig 4b,c authors are referring to Fig 3 b,c
4. In Fig 4a, it would help to add a legend with the color codes.

Reviewer #3 (Remarks to the Author):

This manuscript describes the fabrication of porous nanocomposite scaffolds and the comparison of mineral deposition from human mesenchymal stem cells (hMSCs) cultured dynamically or statically on these scaffolds. The authors use a modified solid casting particulate leaching (SCPL) method to fabricate PLGA nanocomposites containing hydroxyapatite nanoparticles. Quasi-static compression testing was performed to determine the compressive modulus and strength of the scaffolds. hMSCs were cultured under static and dynamic conditions on the scaffolds and bone volume assessed by microCT. Increased mineral deposition and more mature mineral was observed on the dynamic culture samples. These findings are interesting but there are some points that should be addressed. Specific comments:

1. It seems as though the loading platen loses contact with the dynamically loaded scaffolds for half of the loading cycle (negative displacement is associated with no applied force), and it is unclear as to whether the differences between the 1 Hz and 5 Hz loading conditions are significantly different.
2. There appears to be a lot of promise with the micro-CT-based methods used to longitudinally quantify mineral deposition. Grouping bone volume changes by mg HA/cm³ bin is an interesting method for differentiating changes in mature and immature mineral, and using image registration is an excellent method for visualizing regions of mineral deposition and scaffold loss. However, the authors do not relate cell activity or location to bone volume changes or mineral deposition rate. The authors also pool samples with different loading and seeding conditions and do not track hMSC differentiation or look further into what cyclic loading does to the hMSCs.
3. The study design lacks no-cell controls required for confirming cell-mediated mineral deposition.
4. Results and discussion: Nanoparticle reinforced polymer nanocomposite scaffolds. Samples are typically soaked for 24 h prior to quasi-static compression testing, which would allow for comparison to published values and more accurately represent in vivo conditions.
5. Instead of stating that the reference PLGA based scaffolds are "similar," please clarify whether the reference samples are fabricated using the standard SCPL method.
6. The authors cite the agglomeration size from Misra, S.K. et al. (2008). Quantifying the size of the agglomerates in the scaffolds would be more valuable than sharing the SEM image in Fig. 1b.
7. Results and discussion – Monitoring scaffolds in dynamic compression bioreactors. The authors should state whether there are any statistically significant differences.
8. The force vs. time plot in Fig. 2d shows a 0 N force for half of the loading cycle, which suggests that the loading platen is losing contact with the scaffold. If the scaffolds are only subjected to half of the loading sequence, the compressive strains should be 1.5% and 2.5% instead of 3% and 5%.
9. The authors should discuss why the open circle and star exhibit a >100 change of max force (%) before trending with the other groups and why the closed star exhibits a steady scaffold height change (%) and an increase in compressive modulus.
10. Results and discussion – Cell distribution, time-lapsed micro-CT imaging and longitudinal monitoring. The authors should discuss the DNA content differences between the static and dynamic groups.
11. No-cell controls should be included to confirm that the probability density shifts in Fig. 3h-i are cell mediated and not an artifact of soaking the scaffolds in cell culture medium or compressing the scaffolds
12. The rationale for pooling samples in Fig. 3f was not clear, since the loading conditions and seeding densities are different.
13. The authors cite reference thresholds of 97.5 and 130 mg HA/cm³, but they do not specify the significance of these thresholds (e.g. immature mineralized ECM). The authors should also clarify why they selected a density value larger than the reference values.

14. The terms "blackish" and "whitish" are not very specific. Arrows or a legend should be added to Fig. 3d-g.
15. Statistically significant differences in Fig. 3h-j should be noted.
16. Results and discussion – Micro-CT monitoring, image registration and local mineral formation analysis. No cell controls should be included to quantify degradation, deformation, and registration error.
17. Abstract: Some groups apply dynamic fluid flow, so it may be worthwhile to specify dynamic compression instead of "relevant mechanical stimuli." The individual contributions from mechanical stimuli (e.g. fluid flow, hydrostatic pressure, strain, etc.) in vivo aren't very clear.
18. Results and discussion – Monitoring scaffolds in dynamic compression bioreactors. The referenced "30% height reduction" from Baumgartner, W. et al. (2015) should include the loading strain and frequency and should clarify whether samples were dry or wet.
19. Results and discussion – Cell distribution, time-lapsed micro-CT imaging and longitudinal monitoring. The rationale for selection of the 1 Hz, 5% strain group should be described.
20. Results and discussion – Micro-CT monitoring, image registration and local mineral formation analysis. Other groups who have stretched their micro-CT scans before image registration should be cited. The dynamic compression may only affect or damage specific regions of the scaffolds, so stretching the scans may introduce inaccuracies.
21. The rationale for pooling static and dynamic samples when quantifying the linearity of the mineral formation ("n=14") should be explained.
22. Methods – In vitro cell culture. It is difficult to envision how the scaffolds are positioned within the culture chambers. The authors could potentially change Fig. 2a to better show how the chambers and scaffolds fit into the loading/imaging configuration (FIG. 2. in Hagenmuller, et al. (2010)
23. The media flow rate(s) and the frequency of media changes should be specified.

Dear Editors and Reviewers:

Thank you for your letter and for the reviewers' comments concerning our manuscript entitled "Time-lapsed microstructural imaging of polymer nanocomposite scaffolds in dynamic compression bioreactors reveals increased bone formation and mineralization under cyclic loading" (COMMSBIO-20-0533-T). The comments are valuable and helpful for revising and improving our manuscript. We have studied the comments carefully and added new figures to the main manuscript (Fig. R1 and R2, respectively Fig. 5 and 6 in the revised manuscript) and to the supplementary information (Fig. S3-S7). We hope the substantial revisions will be met with approval. Please find below a point-by-point response to the reviewers' comments (*italicized*). Changes to the manuscript are highlighted in yellow.

Thank you very much for your evaluation.

Yours sincerely,

Ralph Müller

On behalf of all contributors

Reviewers' comments:

Reviewer #1 (Remarks to the Author):

This is well designed and performed project. Time-lapsed microCT imaging of cultured scaffolds is innovative. There are no major concerns about the methods and results. But, it would be more interesting to show if the dynamic loading enhanced osteoblast number in the scaffold compared to the static culture.

Our reply: We thank the reviewer for his positive feedback. As part of the major revisions we have added an additional experiment where we compared barium titanate/hydroxyapatite (B3H7) to hydroxyapatite (HA) polymer nanocomposite scaffolds both under dynamic loading and static culture. We have briefly introduced the material in the Introduction (p. 3):

We showed that this bioreactor approach enabled comparison between scaffolds containing pure HA and a mixture of HA and barium titanate (BT). The piezoelectric BT is attractive for bone repair due to its ability to deliver additional electric stimulation³².

For this experiment, we provide endpoint Picro Sirius Red staining with polarized light microscopy for collagen imaging, Alizarin Red S for mineral nodule deposition, and immunohistochemical osteocalcin images to confirm differentiation into osteoblasts (Fig. R1, new Fig. 6). We address the reviewer's comment regarding cell number in the new section "Comparison of hydroxyapatite and barium titanate/hydroxyapatite scaffold materials" (Results and discussion p. 18-19):

'After the culture, we examined those scaffolds by histology. Picro Sirius Red staining showed a distinctively denser collagenous ECM under dynamic conditions compared to static condition, both in B3H7 and HA scaffolds (Fig. 6 a, c vs. b, d). Interestingly, birefringence of collagen fibers, which indicates thicker collagen fibrils⁵⁶ was only observed when the scaffolds were cultured under cyclic loading conditions (Fig. 6 e, g, vs. f, h). The smaller DNA content in loaded scaffolds (Fig. 3a) and collagen staining results suggest that cyclic loading stimulated collagen secretion by osteoblasts but did not result in increased cell number. We then confirmed mineralization by Alizarin Red S staining (Fig. 6 i-l) in subsequent sections. The mineralized ECM was stained (red) in all scaffolds and co-localized with the collagen staining, confirming cell-mediated mineralization. In accordance with the SMD maturation analysis (Fig. 5d), in the B3H7 scaffolds some pores (arrows, Figs. 6i, j) showed intensely stained mineral clusters that were larger in scaffolds cultured under dynamic (Fig. 6i) than static (Fig. 6j) conditions. In addition, B3H7 scaffolds enhanced formation of mineral clusters as compared to HA scaffolds either under static or dynamic conditions. Immunohistochemistry confirmed the presence of osteocalcin, a biochemical marker for osteoblasts in all scaffolds (Fig 6m-p). In line with the SMD maturation analysis and mineral nodule formation observed from Alizarin Red S staining, the amount of osteocalcin appears greater in B3H7 scaffolds than HA scaffolds, irrespective of loading condition.'

Fig. R1 (Fig. 6 in Manuscript) | Extracellular matrix (ECM) products in response to cyclic loading. **a-d**, collagen in red as stained by Picro Sirius Red in B3H7 scaffolds under dynamic, static and HA scaffolds under dynamic as well as static conditions. The scaffold structures were stained brownish to black. **e-h**, Birefringent collagen fibrils of marked regions in **a-d**. **i-l**, Mineralized ECM stained by Alizarin Red. Arrows (**i, j**) point to mineral clusters formed in pores. **m-p**, Immunostaining images of F-actin, nucleus and osteocalcin.

Reviewer #2 (Remarks to the Author):

The manuscript “Time-lapsed microstructural imaging of polymer nanocomposite scaffolds in dynamic compression bioreactors reveals increased bone formation and mineralization under cyclic loading” by Muller et al. present a bioreactor system that combines mechanical loading with longitudinal microCT imaging to assess bone mineralization in a PLGA scaffold reinforced with hydroxyapatite nanoparticles. This novel approach allows for the rapid and rigorous evaluation of engineered bone scaffolds performance in vitro.

Although the use of real time microCT imaging for the evaluation of engineered bone grafts is an exciting approach, the results shown in this manuscript are too preliminary. The authors claim that their approach can identify good candidates for bone replacement, but this is not shown in the manuscript so far.

Our reply: We agree with the reviewer comments that the data does not demonstrate good candidates for bone replacement but rather opens a new venue for the development of potential scaffolds for bone repair. To address this, we have modified the last sentence of the abstract:

‘Therefore, by combining mechanical loading and time-lapsed imaging, this in vitro bioreactor strategy may potentially accelerate development of engineered bone scaffolds and reduce the use of animals for experimentation.’

We have also modified the following sentence to clarify this in the revised manuscript (p. 2, l. 19):

‘In contrast to bone replacement, for bone repair under mechanical load, a scaffold does not require mechanical stiffness and strength as high as dense bone (elastic modulus¹³ = 10-30 GPa) since stabilization is established by fixation.’

The authors study only one type of scaffold, and the increased bone formation due to cyclic compression measured by microCT is not validated with any other techniques that demonstrate that this correlates with better performance.

Our reply: We thank the reviewer for this comment. We have added the following sentence in the manuscript as part of our major revisions (see our next reply) to address concerns regarding validation of micro-CT results (p. 16, l. 18):

Time-lapsed micro-CT imaging allowed to track individual cell-seeded scaffolds and observe their response to changing culture conditions. Previously, it was shown that micro-CT data correlates ($R^2 = 0.96$) with calcium assay levels⁵¹.

Addressing some of the scaffold weaknesses pointed out by the authors (low compressive strengths, low mineral density after culture, weak cell attachment, uneven cell distribution) and/or comparing the performance of different scaffolds in their bioreactor system would demonstrate the potential of this approach and strengthen the relevance of this manuscript.

Our reply: We agree with the reviewer that comparing the performance of different scaffolds in the bioreactors would increase the quality on the manuscript. Therefore, we have included a new Figure 5 (see Figure R.2), as well as Figs. S6, S7 to the supplementary information and added a new section to compare B3H7 and HA scaffolds during an 8 week bioreactor culture.

Results and discussion (p. 15-18) under ‘Comparison of hydroxyapatite and barium titanate/hydroxyapatite scaffold materials’:

‘We next evaluated if time-lapsed micro-CT also allows to compare different bone scaffolds, e.g. scaffolds containing a BT and HA mixture. The piezoelectric property of BT makes it attractive for bone repair due to its ability to deliver additional electric stimulation under cyclic loading. However, the high linear absorption properties of BT (Fig. S6) challenges assessment and comparison to widely used HA based scaffolds by end-point micro-CT. The scaffolds were first cultured for 4 weeks in control medium (CM). Afterwards, the cells were cultured in osteogenic medium (OM) until week 8. For the analysis of the micro-CT scans, a volume of interest in the top 1 mm region of the scaffold was chosen because during the culture in CM air bubbles were observed mainly in the bottom part of the scaffold (Fig. S7, a-d). As the histogram also revealed (Fig. S7 k, l) the presence of air bubbles in the top 1 mm region in the scan from week 1, only scans after week 2 were considered for the analysis.

The same global threshold of 150 mg HA/cm³ was applied for both materials. Figure 5a shows the total BV for B3H7 and HA scaffolds. During the culture in CM, only HA scaffolds exhibited an increase in the total BV that became more distinct in OM. Analysis of the BV growth rate (Fig. 5b), which is similar to the BFR (Fig. 4c) due to negligible BRR (Fig. 4d), shows that HA scaffolds had already a significantly higher rate than B3H7 scaffolds during the culture in CM. Regardless of the scaffold material composition, however, the BV growth rates of scaffolds under dynamic conditions were almost identical to those cultured under static condition. Switching the culture medium to OM had the highest effect on BV growth rate for HA scaffolds under dynamic conditions. For B3H7 scaffolds, switching to OM had only an effect for scaffolds cultured under static conditions. Time-lapsed micro-CT imaging allowed to track individual cell-seeded scaffolds and observe their response to changing culture conditions. Previously, it was shown that micro-CT data correlates ($R^2 = 0.96$) with calcium assay levels⁵³. Cell-seeded silk scaffolds cultured in CM inhibited spontaneous mineralization⁵³. In this work, HA scaffolds exhibited a measurable increase in BV during culture in control medium but there was no observable difference between dynamic and static culture conditions. Therefore, the height loss (Fig. 2e) due to the cyclic loading had no measurable contribution to the scaffolds, corroborating the fact that the increased BV (Fig. 3k) and BFR (Fig. 4c) are due to cell-mediated mineralization. When the culture medium was switched to being osteogenic, this increase was reproduced for HA scaffolds (Fig. 5b) but this increase was not significant, which can be attributed to the comparatively shorter time in osteogenic medium. When the medium was switched to being osteogenic, the BV growth rate only increased for B3H7 scaffolds cultured under static

condition. Under dynamic conditions, the BV growth rate showed a small decrease (Fig. 5b), which was attributed to the height loss (Fig. 2e). A comparison between HA and B3H7 scaffolds based on BV is somewhat limited because B3H7 scaffolds exhibited right from the beginning more voxels above the threshold (Fig. S7 i, j vs k, l) due to the high absorbing BT (Fig. S6). Thus, any mineralization that could occur inside a voxel would not be captured by the BV analysis if that voxel was already above the threshold. Therefore, we analyzed next the SMD, conserving the density information of a voxel. During the culture in control medium all scaffolds exhibited barely any change in SMD (Fig. 5c, regardless of culture condition). Once the scaffolds were cultured in osteogenic medium, B3H7 scaffolds exhibited a strong SMD maturation (Fig. 5c). The SMD maturation rate increased longitudinally, which was significant for all scaffolds and culture conditions (Fig. 5d). However, B3H7 scaffolds showed the strongest effect on media change. The SMD maturation rate under static condition was more than 2x (not significant) and under dynamic more than 5x (significant, $P < 0.001$) higher than for HA scaffolds under dynamic condition.

Fig. R2 (Fig. 5 in Manuscript) Mineral formation kinetics in B3H7 and HA scaffolds. a, Total bone volume (BV) for B3H7 and HA scaffolds under dynamic and static conditions ($n = 5$ for each group). On day 30 (vertical dashed line) the culture medium was switched from control medium (CM) to osteogenic medium (OM). **b,** BV growth rates during culture in CM and OM obtained from linear fits of the data points in **a**. **c,** Scaffold mineral density (SMD) maturation. **d,** the corresponding rates. * $P < 0.05$, ** $P < 0.01$, *** $P < 0.001$; longitudinal groups were tested with paired t-test, one-way

analysis of variance (ANOVA) with post-hoc Bonferroni correction to compare multiple groups separated by CM or OM. In **b** for CM, the groups HA dynamic/static are significantly different from B3H7 dynamic/static. Shapiro-Wilk normality test was not significant.

The production of these scaffolds and a surface functionalization procedure to improve cell adhesion are reported in the Methods section:

‘The process was further upscaled to produce long scaffolds in ferritic nitrocarburized steel molds of 40 mm height, 6 mm inner diameter (X153CrMoV12, Thyssenkrupp). Chloroform (C2432, Sigma-Aldrich) was used to dissolve PLGA of higher inherent viscosity PLGA (Resomer® LG 855 S, Evonik). Consistent to before, the composite was mixed until a homogeneous and viscous mixture was obtained. These mixtures contained less solvent before molding, the porogen content was increased to result in HA nanocomposites with a 1:1:12 wt. ratio between PLGA, nanofiller and NaCl. Thus, the mass loss after porogen leaching was held constant at ca. $88.6 \pm 1.4\%$, consistent to before. The solid filler volume fraction in scaffolds containing BT and HA was held constant at ca. 30 vol%.’

‘**Surface functionalization.** Scaffolds that received surface functionalization were immersed in an aqueous 1 wt% Poly(vinyl alcohol) solution (22225, MW 6000, 80% hydrolyzed, Polysciences Inc.) for 15 min at the end of the salt leaching process to render the surface hydrophilic⁵⁹. After H₂O₂ plasma sterilization, scaffolds were surface functionalized⁶⁰ by pipetting 40 µl of 1 mg/ml Arginylglycylaspartic acid peptide motif (4008998, GRGDS, Bachem) and 2 mg/ml dopamine hydrochloride (H8502, Sigma) in 10 mM, pH 8.5 Tris buffer (T6791, Sigma-Aldrich). The scaffolds were incubated for 2 h at 37°C and then washed 3x with PBS and then stored in an incubator (37 °C, 5% CO₂) until cell seeding.’

1. *Supplementary Fig 1 and 2 were not provided to the reviewers.*

Our reply: We apologize to have missed to include the supplementary information in the transfer process. We made sure to include the supplementary information in the resubmission.

2. *Authors mention how traditional in vitro assays evaluate bone engineered scaffolds based on markers of osteogenesis and end-point calcium levels. It would be interesting to see those in this paper to be able to compare with previous studies and see how the osteogenesis markers and calcium levels correlate with the increased bone density measured in this study. Similarly, further histological analysis of the samples to show new bone formation and colocalization with cell clusters will strengthen the results.*

Our reply: We thank the reviewer for the suggestions to further strengthen our manuscript. We have incorporated the following sentence (p.16, l. 19) in the new section ‘Comparison of hydroxyapatite and barium titanate/hydroxyapatite scaffold materials’:

‘Previously it was shown that micro-CT data correlates ($R^2 = 0.96$) with end-point calcium assay levels⁵³.’

Additionally, we provided histological images in the new Fig. 6 (see Fig. R1) of samples stained with Alizarin Red S for minerals and Picro Sirius Red for collagen, as well as osteocalcin as osteogenesis marker to corroborate the results reported by micro-CT. We added on p. 18-19 (same response as to reviewer #1):

‘After the culture, we examined those scaffolds by histology. Picro Sirius Red staining showed a distinctively denser collagenous ECM under dynamic conditions compared to static condition, both in B3H7 and HA scaffolds (Fig. 6 a, c vs. b, d). Interestingly, birefringence of collagen fibers, which indicates thicker collagen fibrills⁵⁶ was only observed when the scaffolds were cultured under cyclic loading conditions (Fig. 6 e, g, vs. f, h). The smaller DNA content in loaded scaffolds (Fig. 3a) and collagen staining results suggest that cyclic loading stimulated collagen secretion by osteoblasts but did not result in increased cell number. We then confirmed mineralization by Alizarin Red S staining (Fig. 6 i-l) in subsequent sections. The mineralized ECM was stained (red) in all scaffolds and co-localized with the collagen staining, confirming cell-mediated mineralization. In accordance with the SMD maturation analysis (Fig. 5d), in the B3H7 scaffolds some pores (arrows, Figs. 6i, j) showed intensely stained mineral clusters that were larger in scaffolds cultured under dynamic (Fig. 6i) than static (Fig. 6j) conditions. In addition, B3H7 scaffolds enhanced formation of mineral clusters as compared to HA scaffolds either under static or dynamic conditions. Immunohistochemistry confirmed the presence of osteocalcin, a biochemical marker for osteoblasts in all scaffolds (Fig 6m-p). In line with the SMD maturation analysis and mineral nodule formation observed from Alizarin Red S staining, the amount of osteocalcin appears greater in B3H7 scaffolds than HA scaffolds, irrespective of loading condition.’

3. *The authors explain that the mineral density formed in these scaffolds under cyclic loading is still well below mineralized tissue formation during in vivo defect healing. What’s the authors hypothesis to explain this? What do these differences between in vivo and in vitro behavior mean in terms of the value of this bioreactor testing strategy?*

Our reply: This is an intriguing question – thank you. We first want to note that also in vivo, depending on the defect size, there are differences in the mineral density formed. Thus, we modified the manuscript (p. 13) and specified that:

‘This in vitro formed mineral density is consistent with mineral formation in silk scaffolds⁴⁸, however, it is still well below mineralized tissue formation during non-critical-sized in vivo defect healing³⁰.’

Additionally, we have added the following sentence in the results and discussion part (p. 19) to provide a hypothesis on the differences in the formed mineral density:

‘In this in vitro study the density of formed mineralized tissue in HA scaffolds did not reach in vivo values. On the other hand, in critical-sized in vivo bone defects, polymer nanocomposite scaffolds containing tricalcium phosphate⁵ or hydroxyapatite⁶ particles were without bone morphogenetic proteins also not able to restore the defect with higher density bone ($> 500 \text{ mg HA/cm}^3$). Thus, we believe the formed mineralized tissue of lower density is due to the insufficient osteoinductive properties of HA scaffolds. The B3H7 scaffolds exhibited a significantly enhanced SMD maturation rate (Fig. 6d) which is likely due to the formation of mineral clusters in pores. Time-lapsed micro-CT imaging of scaffolds in dynamic compression bioreactors is therefore an effective strategy to identify promising candidates for bone tissue engineering applications.’

Minor comments

1. *Based on the methods I understand that the scaffolds analyzed in figure 2 were seeded with bone marrow stromal cells, but not for the analysis performed in Fig 1. This should be made clear in the text.*

Our reply: We thank the reviewer for their comment. We have made the following changes to clarify that scaffolds were dry and unseeded (Fig. 1 caption):

‘The compressive stress as a function of strain of dry and unseeded HA55-PLGA (u) with a 2:1 height to diameter aspect ratio (solid red line, $n = 5$), and scaffolds made by standard SCPL²⁵ with aspect ratio $< 1:1$ (blue dotted line, $n = 1$) or 2:1 (green broken line, $n = 2$).’

On p.5, l. 3: ‘Figure 1c shows compressive stress as a function of strain of dry and unseeded nanocomposite scaffolds with 2:1 and $< 1:1$ height to diameter aspect ratios.’

On p. 6, l. 12: ‘We compared the mechanical properties of these dry and unseeded scaffolds (Figure 1d: cross, star) with published PLGA based dry bone scaffolds that have porosities larger than 80% to shed light on the influence of filler vol%, SSA and processing.’

We also modified the section ‘Monitoring scaffolds in dynamic compression bioreactors’ on p. 7 in the manuscript:

‘For the bioreactor culture, scaffolds were seeded with human marrow stromal cells (hMSCs).’

2. It is not clear what the Fig 1b zoom in is trying to show. Adding some arrows or clarifications might help

Our reply: We thank the reviewer for the comment. To clarify what this, we have modified the figure caption 1b as follows:

‘3D model obtained from that micro-CT scan, along with a scanning electron microscopy image of the surface showing HA nanoparticles exposed at the surface (arrows).’

3. In Page 14 Line 1, where it says Fig 4b,c authors are referring to Fig 3 b,c

Our reply: We thank the reviewer for noticing the wrong figure labeling. We have corrected the figure numbers accordingly.

4. In Fig 4a, it would help to add a legend with the color codes.

Our reply: We agree with the reviewer comment. We have included the legend with the color codes in Figure 4a.

Reviewer #3 (Remarks to the Author):

This manuscript describes the fabrication of porous nanocomposite scaffolds and the comparison of mineral deposition from human mesenchymal stem cells (hMSCs) cultured dynamically or statically on these scaffolds. The authors use a modified solid casting particulate leaching (SCPL) method to fabricate PLGA nanocomposites containing hydroxyapatite nanoparticles. Quasi-static compression testing was performed to determine the compressive modulus and strength of the scaffolds. hMSCs were cultured under static and dynamic conditions on the scaffolds and bone volume assessed by microCT. Increased mineral deposition and more mature mineral was observed on the dynamic culture samples. These findings are interesting but there are some points that should be addressed.

Specific comments:

1. *It seems as though the loading platen loses contact with the dynamically loaded scaffolds for half of the loading cycle (negative displacement is associated with no applied force), ...*

Our reply: We thank the reviewer for the comment. It is correct that the loading platen is losing contact with the scaffold during half of the loading cycle. We modified the text (p. 7) to clarify this in the manuscript:

‘The MSU allowed controlling the force F_{thres} to contact the scaffold, as well as frequency and strain (Fig. 2a) of the loading regime. One scaffold was cultured under static conditions and the other was loaded cyclically 3 times per week for 5 min. First needle-like HA55-PLGA (u) nanocomposite scaffolds have been chosen because the employed nanoparticles are established in literature²⁵. Before loading, the scaffolds were contacted using $F_{\text{thres}} = 0.05$ N, so that during the full displacement (Fig. 2c, ca. 180 μm) the specified target strain (Fig. 2c, 5%) was reached, resulting in a peak of the force measurement (Fig. 2d). Therefore, during half of the loading cycle, the piston was not in contact with the scaffold (Fig. 2d, 0 N force). Using $F_{\text{thres}} = 0.05$ N, scaffolds were loaded first with a loading scenario of 1 Hz and 5% strain and then in a second independent experiment with 5 Hz and 3% strain.’

... and it is unclear as to whether the differences between the 1 Hz and 5 Hz loading conditions are significantly different.

Our reply: We understand that in the original manuscript it was confusing to compare both loading conditions in a single figure without conducting a statistical test. However, the selection of the different loading conditions were a heuristic optimization approach based on the mechanical monitoring data. To clarify this, we have added a new Fig. S3 in the supplementary information and rearranged Fig. 2 (see Fig. R3 in this reply) by splitting the different loading regimes in Fig. 2 into sub figures. As we do not directly compare the different loading conditions in a single figure anymore, we specify statistical significance only in the text. The ‘Monitoring scaffolds in dynamic compression bioreactors’ section under the Results and discussion section (p. 7-9) has been modified as follows:

‘The influence of cyclic compression to changes in scaffold height and max. force was further investigated in different scaffold compositions. We compared a mixture (3:7 vol. ratio) of BT and HA scaffolds (B3H7) to HA20-PLGA (u) scaffolds, denoted as HA scaffolds. The BT and HA nanoparticles were of comparable size and spherical shape to exclude any potential size or shape effects. Before the compression bioreactor culture, all produced scaffolds were first tested dry and unseeded using nondestructive stress-strain measurements. The analysis of the stress-strain measurement using $F_{\text{thres}} = 0.05$ N (Fig. S3a) showed that the scaffold was contacted in the toe region (Fig. S3a). Therefore, we chose a higher $F_{\text{thres}} = 0.2$ N to measure the compressive modulus (Fig. S3b). For the compression bioreactor culture, scaffolds were selected to have an equal compressive modulus across all groups (Fig. S3c).

Typical for polymer foams⁴¹, the force response decayed rapidly during the first few compression cycles (Fig. 2c) and then in a second phase decreased very slowly (Fig. 2d, representative sample), showing that the mechanical integrity of the scaffold was maintained during the culture. On the other hand, each scaffold (individual symbols in Figs. 2e, f, left and middle) typically lost height over the course of the cell culture (Fig. 2e). This height loss was a bit more pronounced (not significant) with the loading scenario 1 Hz, 5% strain (Fig. 2e, left) compared to the 5 Hz, 3% strain (Fig. 2e, middle, right) and is attributed to the regular cyclic loading. The height reduction of up to 17% (Fig. 2e, left, circle) after 7 weeks of cyclic loading (3 times per week of 5 min long) was still smaller than for cell-seeded electrospun calcium phosphate-PLGA nanocomposite scaffolds¹², which showed a 30% height reduction after nine days of daily cyclically loading for 10 min with 1 Hz and 5% strain. On the last day of the experiment B3H7 scaffolds cultured under 5 Hz, 3% strain showed a significantly lower ($p < 0.05$) height reduction than HA scaffolds cultured under 1 Hz, 5% strain. Figure 2f shows the change of the max. force response over the course of the culture. For scaffolds contacted with $F_{\text{thres}} = 0.05$ N (Fig. 2f, left, middle), the max. force typically decreased with time until after 3-4 weeks it remained relatively stable for each scaffold (individual symbols). In contrast to the scaffold height, scaffolds loaded with 5 Hz, 3% strain (Fig. 2f, middle) exhibited lower max. force values. A $F_{\text{thres}} = 0.2$ N contact force resulted in relatively stable max. force values throughout the experiment while it did not affect the change in scaffold height. Therefore, variations of samples observed in Fig. 2e-middle and Fig. 2f-left (triangle, pentagon) are attributed to a F_{thres} that is not adequately adjusted to the scaffold's mechanical properties. Additionally, such variations will be increased by scaffold surfaces parallel to the piston not being perfectly plane and aligned.'

Fig. R3 (Figure 2 in Manuscript) | Monitoring of cell-seeded scaffolds in dynamic compression bioreactors. **a**, Scheme of two scaffolds fixed inside the bioreactor placed within the mechanical stimulation unit. The contact force (F_{thres}), frequency and strain can be controlled. The red scaffold is loaded cyclically. **b**, Raw micro-CT image of the bioreactors showing the two scaffolds. **c**, Displacement recording during cyclic compression with 1 Hz, and 5% strain and **d** the corresponding force recording. **e**, Height change of scaffolds during dynamic culture with 1 Hz, and 5% strain (left), 5 Hz, 3% strain (middle) using $F_{thres} = 0.05$ N; the latter also with $F_{thres} = 0.2$ N (right). **f**, The corresponding change of max. force response. **Symbols with error bar represent the mean and s.d. No significant difference was found between B3H7 and HA.**

2. There appears to be a lot of promise with the micro-CT-based methods used to longitudinally quantify mineral deposition. Grouping bone volume changes by mg HA/cm³ bin is an interesting method for differentiating changes in mature and immature mineral, and using image registration is an excellent method for visualizing regions of mineral deposition and scaffold loss. However, the

authors do not relate cell activity or location to bone volume changes or mineral deposition rate. The authors also pool samples with different loading and seeding conditions and do not track hMSC differentiation or look further into what cyclic loading does to the hMSCs.

Our reply: We thank the reviewer for the constructive feedback. In this new version of the manuscript, we have included a new Fig. 5 (Fig. R2) and new Fig. 6 (Fig. R1) and a new section (p.15-19) ‘Comparison of hydroxyapatite and barium titanate/hydroxyapatite scaffold materials’ under the Results and discussion part to relate cell activity, to scaffold mineral density (SMD) maturation and also look further into what cyclic loading does to the hMSCs. We address the reviewer comments in the following sentences of the new section (same response as to reviewer #1).

‘After the culture, we examined those scaffolds by histology. Picro Sirius Red staining showed a distinctively denser collagenous ECM under dynamic conditions compared to static condition, both in B3H7 and HA scaffolds (Fig. 6 a, c vs. b, d). Interestingly, birefringence of collagen fibers, which indicates thicker collagen fibrils⁵⁶ was only observed when the scaffolds were cultured under cyclic loading conditions (Fig. 6 e, g, vs. f, h). The smaller DNA content in loaded scaffolds (Fig. 3a) and collagen staining results suggest that cyclic loading stimulated collagen secretion by osteoblasts but did not result in increased cell number. We then confirmed cell-mediated mineralization by Alizarin Red S staining (Fig. 6 i-l) in subsequent sections. The mineralized ECM was stained (red) in all scaffolds and co-localized with the collagen staining. In accordance with the SMD maturation analysis (Fig. 5d), in the B3H7 scaffolds some pores (arrows, Figs. 6i, j) showed intensely stained mineral clusters that were larger in scaffolds cultured under dynamic (Fig. 6i) than static (Fig. 6j) conditions. In addition, B3H7 scaffolds enhanced formation of mineral clusters as compared to HA scaffolds either under static or dynamic conditions. Immunohistochemistry confirmed the presence of osteocalcin, a biochemical marker for osteoblasts in all scaffolds (Fig 6m-p). In line with the SMD maturation analysis and mineral nodule formation observed from Alizarin Red S staining, the amount of osteocalcin appears greater in B3H7 scaffolds than HA scaffolds, irrespective of loading condition.’

We have now split the two experiments and apologize for having them pooled without any clear rationale. The results for the second experiment are reported in the supplementary information Fig. S5 (see Fig. R5). We modified the manuscript (p. 13) to refer to the supplementary information:

‘The results presented in Fig. 3 were reproduced for scaffolds cultured with a 5 Hz and 3% strain loading scenario (Figure S5).’

3. The study design lacks no-cell controls required for confirming cell-mediated mineral deposition.

Our reply: We thank the reviewer for this constructive feedback and agree that cell-mediated mineral deposition should be further confirmed. However, we have previously established that in acellular scaffolds, spontaneous mineralization from medium precipitation occurs more frequently than with cells in the culture [Fig. 5 in Vetsch et al. *Acta Biomater.* **13**, 277–285 (2015)], which makes it difficult to use acellular controls. Nevertheless, in order to prove cell-mediated response, immunohistochemistry for osteocalcin and calcium deposition stainings were included in the new Fig. 6 (see Fig. R1). The following sentence was included (p. 19):

‘We then confirmed mineralization by Alizarin Red S staining (Fig. 6 i-l) in subsequent sections. The mineralized ECM was stained (red) in all scaffolds and co-localized with the collagen staining, confirming cell-mediated mineralization. In accordance with the SMD maturation analysis (Fig. 5d), in the B3H7 scaffolds some pores (arrows, Figs. 6i, j) showed intensely stained mineral clusters that were larger in scaffolds cultured under dynamic (Fig. 6i) than static (Fig. 6j) conditions.’

4. Results and discussion: Nanoparticle reinforced polymer nanocomposite scaffolds. Samples are typically soaked for 24 h prior to quasi-static compression testing, which would allow for comparison to published values and more accurately represent in vivo conditions.

Our reply: The reviewer has correctly pointed out that samples should be typically soaked for 24 h prior to quasi-static compression testing as this also represents more accurately in vivo conditions. However, we found more literature that reported mechanical properties of dry scaffolds. Thus, we also used dry scaffolds in Fig. 1 to benchmark our modified SCPL method for producing reinforced nanocomposite scaffolds.

We made the following change to the caption of Fig. 1d:

‘**d**, Compressive moduli and **e** strength of bone¹⁵ or dry and unseeded nanocomposite scaffolds of various filler specific surface areas as a function of filler vol% and similar ones from literature^{25,34,38,39}.’

Furthermore, we also mention the limitation of dry testing in our discussion (p. 7):

‘The dry mechanical tests limit the comparison to in vivo conditions, for which the scaffolds should be typically soaked for 24 h prior to quasi-static compression testing.’

5. Instead of stating that the reference PLGA based scaffolds are “similar,” please clarify whether the reference samples are fabricated using the standard SCPL method.

Our reply: We thank the reviewers for the comment. We have modified the manuscript to clarify the production method of reference samples (p. 6, l. 11):

‘We compared the mechanical properties of these dry and unseeded scaffolds (Figure 1d: cross, star) with published PLGA based dry bone scaffolds that have porosities larger than 80% to shed light on the influence of filler vol%, SSA and processing. The reference scaffolds were prepared by gas-foaming particulate leaching²⁵ (pentagon), high-pressure compression molding³⁴ (triangle-up), SCPL³⁸ (hexagon) or 3D³⁹ bioplotted (circle).’

6. *The authors cite the agglomeration size from Misra, S.K. et al. (2008). Quantifying the size of the agglomerates in the scaffolds would be more valuable than sharing the SEM image in Fig. 1b.*

Our reply: Thank you for your valuable suggestion, which we implemented in the following way. The hydrodynamic agglomerate size was measured by dynamic light scattering using a Zetasizer (Nano ZS, Malvern Instruments) right after deagglomeration by ultrasonication (Sonics) and is now reported in Table 1. We also like to note that Misra, S.K. et al. (2008) reports the influence of agglomerate size on mechanical properties, the 6.5 μm agglomerates is from Zhang et al. (2016).

Following changes were made to the manuscript (p. 6, l. 23): ‘This inferior property is likely due to the much larger fillers³³ with 6.5 μm agglomerates³⁴ compared to the sub-micron sized agglomerates used in this study (Table 1), lower filler vol% and lack of proper dispersion of nanoparticles in the polymer matrix.’

Methods (p. 20): ‘The hydrodynamic agglomerate size was measured by dynamic light scattering using a Zetasizer (Nano ZS, Malvern Instruments) right after deagglomeration by ultrasonication (Sonics).’

0. *Results and discussion – Monitoring scaffolds in dynamic compression bioreactors. The authors should state whether there are any statistically significant differences.*

Our reply: As part of the major revisions we changed Fig. 2. We modified the caption of Fig. 2 and the text to mentioned whether there are any statistically significant differences.

Caption of Fig. 2 (see Fig. R3): ‘e, Height change of scaffolds during dynamic culture with 1 Hz, and 5% strain (left), 5 Hz, 3% strain (middle), both with $F_{\text{thres}} = 0.05 \text{ N}$, the latter also with $F_{\text{thres}} = 0.2 \text{ N}$ (right). f, The corresponding change of max. force response. No significant difference was found between B3H7 and HA scaffolds.’

On p. 9, l. 8: 'On the last day of the experiment B3H7 scaffolds cultured under 5 Hz, 3% strain showed a significantly lower ($p < 0.05$) height reduction than HA scaffolds cultured under 1 Hz, 5% strain.'

8. *The force vs. time plot in Fig. 2d shows a 0 N force for half of the loading cycle, which suggests that the loading platen is losing contact with the scaffold. If the scaffolds are only subjected to half of the loading sequence, the compressive strains should be 1.5% and 2.5% instead of 3% and 5%.*

Our reply: It is correct that the loading platen is losing contact with the scaffold during half of the loading cycle. However, the reviewer took wrong conclusions from Fig. 2d. We modified the manuscript on p. 7, line 20 to clarify that:

Before loading, the scaffolds were contacted using $F_{\text{thres}} = 0.05$ N, so that during the full displacement (Fig. 2c, ca. 180 μm) the specified target strain (Fig. 2c, 5%) was reached, resulting in a peak of the force measurement (Fig. 2d). Therefore, during half of the loading cycle, the piston is not in contact with the scaffold (Fig. 2d, 0 N force). Using $F_{\text{thres}} = 0.05$ N, scaffolds were loaded [...]

9. *The authors should discuss why the open circle and star exhibit a >100 change of max force (%) before trending with the other groups and why the closed star exhibits a steady scaffold height change (%) and an increase in compressive modulus.*

Our reply: We thank reviewer for the detailed inspection of our data. As already mentioned in the manuscript (initial submission p. 8, l. 3-6): 'The variations observed in Fig. 2e-g are due to measurement uncertainty of contacting the scaffolds with the piston using a small threshold force of 0.05 N and scaffold surfaces parallel to the piston being not perfectly plane or aligned.'

We added new B3H7 (see also response to reviewer 2) and additional HA polymer nanocomposite scaffolds data to corroborate this statement. Changes in the manuscript were mentioned in our response to your comment 1. During these measurements we also checked again the force, strain measurements of the mechanical stimulation unit. We realized that the machine stiffness of the bioreactor and loading device system is similar to our scaffold. Because the machine compliance was also not linear, accurate correction is not possible. Therefore, we regretfully removed the compressive modulus data from Fig. 2 and its discussion.

0. *Results and discussion – Cell distribution, time-lapsed micro-CT imaging and longitudinal monitoring. They authors should discuss the DNA content differences between the static and dynamic groups.*

Our reply: We thank the reviewer to bring the missing discussion to our attention. We have added the following discussion on p. 10 under ‘Cell distribution, time-lapsed micro-CT imaging and longitudinal monitoring’:

‘Under dynamic culture conditions there was a significant smaller DNA amount compared to static condition. This additional decrease may have resulted from enhanced mineralization as a fraction of the osteoblasts undergo apoptosis after completing their bone-forming function⁴⁶.’

11. *No-cell controls should be included to confirm that the probability density shifts in Fig. 3h-i are cell mediated and not an artifact of soaking the scaffolds in cell culture medium or compressing the scaffolds.*

Our reply: We partly disagree with the reviewer. We have previously established that in acellular scaffolds, spontaneous mineralization from medium precipitation occurs more frequently than with cells in the culture [Fig. 5 in Vetsch et al. *Acta Biomater.* **13**, 277–285 (2015)], which makes it difficult to use acellular controls. The reviewer is right, soaking the scaffolds in cell culture medium would certainly cause such a shift. Pores that are not filled with cell culture medium, i.e. they contain air bubbles, display mineral density values below zero (please see also new Fig. S7 for the new data that was added). Therefore, such regions must be excluded for the analysis. We have therefore used scans from week 2 as reference to ensure that the scaffolds were completely soaked with medium, and write on p. 12:

‘Images from week 2 were chosen as reference because at that time-point the scaffolds were completely soaked with medium.’

We have added the following information to the supplementary information regarding the reviewer’s concern of shifting the density distribution by compressing an acellular scaffold:

‘To rule out that compressing the scaffolds would cause the shift, we first fully soaked a scaffold by immersion in Dulbecco’s Modified Eagle Medium (DMEM) under vacuum. A first micro-CT scan and a histogram of the image proved that the scaffold was fully soaked with DMEM. Then, we applied 3 consecutive cyclic loadings and took again a micro-CT scan. In total, this procedure was repeated 5 times to simulate 5 weeks of normal culture while at the same time avoiding any mineral precipitation. The scaffold height dropped from the first to the last scan by 6.4%. Figure R2 shows that the probability density showed no shift but only a widening of the distribution. The observed changes are within the standard deviation of the distributions shown in Fig. 3h-i.’

Fig. R4 (Figure S4 in the Supplementary Information) | Probability density distributions of compressed scaffolds. a, the change of the probability distribution of a scaffold that was compressed by applying repeatedly 3 consecutive loadings with 5 Hz, 3% strain using $F_{\text{thres}} = 0.2 \text{ N}$. **b,** vertical micro-CT cross section of the scaffold at 100% height and **c,** after the last loading at 93.6% height.

In the manuscript we note on p. 12, l. 23: ‘Mere scaffold compression results in a negligible widening of the density distribution (Figure S4).’

12. *The rationale for pooling samples in Fig. 3f was not clear, since the loading conditions and seeding densities are different.*

Our reply: We apologize for pooling the results without a clear rationale. Originally, we pooled the results because the change of total BV was reproduced for both loading scenarios and no significant difference was found. Furthermore, previous studies have suggested that osteogenesis might be proportional to the equation $\epsilon^2 \log(2)$, where ϵ is peak applied strain and f is frequency [Hsieh, Y.-F. & Turner, C. H. *J. Bone Miner. Res.* **16**, 918–924 (2001)]. Thus, the loading conditions (5 % strain, 1 Hz vs. 3% strain, 5 Hz) are not that different. Nevertheless, also because of our response to the reviewer’s comment 19, we decided to present the data separately. We present the results of the 5 Hz and 3% in Fig. S5 (see Fig. R5) and modified the manuscript on p. 13:

‘The results presented in Fig. 3 were reproduced for scaffolds cultured with a 5 Hz and 3% strain loading scenario (Figure S5).’

13. *The authors cite reference thresholds of 97.5 and 130 mg HA/cm³, but they do not specify the significance of these thresholds (e.g. immature mineralized ECM). The authors should also clarify why they selected a density value larger than the reference values.*

Our reply: We clarified the selection of our threshold value in the manuscript on p. 13, l. 1. : ‘These thresholds were chosen to distinguish mineralized ECM from the background, e.g. culture medium, and corresponded to small mineral nodules⁸. Here, a slightly higher threshold was chosen to reduce

partial volume effects because the scaffolds were already pre-mineralized due to the embedded nanoparticles.'

Fig. R5 (Figure S5 in the Supplementary Information) | Time-lapsed micro-CT monitoring of HA55-PLGA (u) scaffolds, 5 Hz and 3% strain. a, Micro-CT cross-sectional slice of a scaffold cultured under static condition from Week 2 and b Week 6, respectively of a scaffold cultured under dynamic condition (c-d). Histogram of such images from static (e, n = 4) and dynamic (f, n = 2) cultured scaffolds. g, BV for scaffold mineral density (SMD) 150 and 175 mg HA/cm³, respectively h for SMD bins 200-450. The bin width is 25 mg HA/cm³. i, Total BV as function of time for static (n = 4) and dynamic (n = 3 in week 2-5, n = 2, Week 6) culture conditions. Symbols represent the mean

and error bars the s.d. (*P < 0.05); t-tests were used to highlight significant differences between both conditions for each time-point. Shapiro-Wilk and Levene's Test were not significant.

14. The terms “blackish” and “whitish” are not very specific. Arrows or a legend should be added to Fig. 3d-g.

Our reply: We thank the reviewer to help improving the clarity of our figure. We added a white arrow pointing to the “blackish” pixels and a yellow arrow pointing to the “whitish” pixels in Fig. 3d-g.

15. Statistically significant differences in Fig. 3h-j should be noted.

Our reply: We agree with the reviewer and therefore did paired Student's t-test to compare the areas below the distributions for HA concentrations above the threshold (150 mg HA/cm³) in Fig. 3h-i.

The manuscript was modified as follows (p. 12): ‘The probability for voxels exhibiting a mineral density > 150 mg HA/cm³ was significantly larger at week 7 compared to week 2 for both static (p < 1E-5) and dynamic (p < 1E-3) conditions.’

We did Student's t-test to compare the BV at week 7 between dynamic and static conditions for each density bin.

The manuscript was modified as follows (p. 13): ‘The differences in BV between the dynamic and static condition were statistically significant at week 7 for all mineralization levels up to 250 mg HA/cm³, where also most of the mineral formation (BV > 0.1 mm³) occurred.’

16. Results and discussion – Micro-CT monitoring, image registration and local mineral formation analysis. No cell controls should be included to quantify degradation, deformation, and registration error.

Our reply: As mentioned previously, we must disagree with the reviewer for the reasons pointed out in our response to comment 11. Acellular scaffolds have exhibited significant mineral precipitation in the past [Fig. 5 in Vetsch et al. *Acta Biomater.* **13**, 277–285 (2015)] and are therefore not suitable to quantify registration errors. Micro-CT imaging of mineralized scaffolds was shown to capture cell-mediated mineral formation with high accuracy (mean percentage difference: 1.4-14%) in comparison with histology [Thimm et al. *Ann. Biomed. Eng.* **41**, 2666–2675 (2013)]. Nevertheless, to clarify potential errors of the image registration we added the following sentences:

Results and discussion (p. 14): ‘Consequently, any changes in the quiescent (grey) sites may also be considered as an error because they should stay constant.’

Methods section (p. 26): ‘Changes in the quiescent volumes were considered as registration error and quantified as the coefficient of variation, which is the standard deviation of the quiescent volumes from the different time-points normalized by the average value of the quiescent volumes.’

and added to the supplementary information.

Table R1 (corresponds to Table S1) | Coefficient of variation (CV) for quiescent volume.

CV (%)	Scaff. 1	Scaff. 2	Scaff. 3	Scaff. 4	Scaff. 5	Scaff. 6	Scaff. 7
top	2.19	1.38	2.40	2.24	0.89	0.94	1.58
bottom	2.16	1.08	1.41	2.00	1.8	1.1	0.28

Results and discussion (p. 14): ‘Note that BRR values were about a tenth of BFR values. Also, the coefficient of variation (CV) of the quiescent volume within the selected volume of interest was for all samples $\leq 2.4\%$ - smaller than the CV = [10.7%, 30.4%] of the reported BFR values. Therefore, any erroneous contribution from scaffold shrinkage, deformation or registration error towards BFR was within the standard deviation of the measurement, and is thus negligibly small. As this was not the case for scaffolds loaded with 1 Hz and 5% strain, they were not considered for image registration.’

17.Abstract: Some groups apply dynamic fluid flow, so it may be worthwhile to specify dynamic compression instead of “relevant mechanical stimuli.” The individual contributions from mechanical stimuli (e.g. fluid flow, hydrostatic pressure, strain, etc.) in vivo aren’t very clear.

Our reply: We agree with the reviewer and we have modified ‘relevant mechanical stimuli’ to ‘dynamic compression’.

The abstract was modified as follows: ‘In contrast to in vivo, in vitro studies are often conducted in the absence of dynamic compression and seldom focus on assessing bone formation and mineralization.’

18.Results and discussion – Monitoring scaffolds in dynamic compression bioreactors. The referenced “30% height reduction” from Baumgartner, W. et al. (2015) should include the loading strain and frequency and should clarify whether samples were dry or wet.

Our reply: We modified the manuscript to clarify that our samples for Fig. 2 were cell-seeded, i.e. wet (see response to minor comment 1 from reviewer 2), and specified on p. 9 that the reference scaffolds were cell-seeded and that 1 Hz and 5% strain was used:

‘The height reduction of up to 17% (Fig. 2e, left, circle) after 7 weeks of cyclic loading (3 times per week of 5 min long) was still smaller than for cell-seeded electrospun calcium phosphate-PLGA nanocomposite scaffolds¹², which showed a 30% height reduction after nine days of daily cyclically loading for 10 min with 1 Hz and 5% strain.’

19. Results and discussion – Cell distribution, time-lapsed micro-CT imaging and longitudinal monitoring. The rationale for selection of the 1 Hz, 5% strain group should be described.

Our reply: We thank the reviewer for his comment. We presented the data for the 1 Hz, 5% strain because we had more samples for that group. For the 5 Hz, 3% strain group one scaffold was detached from the substrate after week 5 (Fig. R1e, middle). We have now carefully inspected our data again and realized that we forgot to note that we had technical issues because of an ageing X-Ray tube. We note this in the methods section on p. 24 of the revised manuscript: ‘Because of a technical issue with an aging X-Ray tube, for the 2nd experiment the bioreactors had to be scanned at 165 μ A. The total energy was kept constant by using 215 ms integration time and a frame averaging of 3.’ Also, no more measurements could be taken after week 6. For these reasons we decided to present the data for the 5 Hz, 3% strain group separately in the supplementary information (Fig. S5). Consistent with 1 Hz, 5% strain, scaffolds cultured under dynamic conditions exhibited a trend towards more mature mineral compared to static culture condition. The following section and Fig. S5 were added in the supplementary information:

‘The scaffold height loss, which was induced by cyclic compression, is barely visible and no obvious difference is detectable by eye between the scaffold cultured under static and dynamic condition (Figure S5a-d). Consistent with Fig. 3h-i, the histograms exhibit a shift in the probability density distributions between the two time-points. Note, that the scaffolds cultured under dynamic conditions (Fig. S5i), have the same height loss as the scaffold in the control experiment (Fig. S4). Nevertheless, they exhibit an obvious increase of the probability density at HA concentrations above the threshold, corroborating the fact that the observed changes in the probability density were not due to height loss. The distribution of the histogram is slightly more skewed towards higher densities (Fig. S5e-f vs. Fig. 3h-i), which can be attributed to the lower current intensity (165 vs. 177 μ A) that had to be chosen due to technical issues with the X-Ray tube of the micro-CT. Consistent with Fig. 3, more mineral matured in scaffolds cultured under dynamic conditions compared to static condition. However, because of the skewed histogram, a difference in BV (10-50% at Week 6) between dynamic and static conditions was

observed for scaffold mineral densities $> 175 \text{ mg HA/cm}^3$ (Fig. S5g-h). Also, the total BV was higher in scaffolds cultured under dynamic conditions.’

20. Results and discussion – Micro-CT monitoring, image registration and local mineral formation analysis. Other groups who have stretched their micro-CT scans before image registration should be cited. The dynamic compression may only affect or damage specific regions of the scaffolds, so stretching the scans may introduce inaccuracies.

Our reply: We have modified the manuscript on p. 14 and cite another paper that uses non-rigid registration:

‘Also, because scaffolds under dynamic conditions are compacted during culture, we applied a manual non-rigid registration⁵² before automated registration, which was accomplished by stretching the scaffolds in later time-points to the height of week 2 assuming a linear deformation. This correction was needed to avoid mistakenly labelling voxels as mineral formation or resorption sites because of scaffold deformation.’

21. The rationale for pooling static and dynamic samples when quantifying the linearity of the mineral formation (“n=14”) should be explained.

Our reply: When calculated separately, the average value remains 0.998 ± 0.003 for each group and region. Thus, for simplicity we pooled them. We modified the manuscript on p. 14 to clarify the rationale of pooling static and dynamic samples:

‘There was no difference between static and dynamic samples regarding linearity, independent of the analyzed region (top or bottom).’

22. Methods – In vitro cell culture. It is difficult to envision how the scaffolds are positioned within the culture chambers. The authors could potentially change Fig. 2a to better show how the chambers and scaffolds fit into the loading/imaging configuration (FIG. 2. in Hagenmuller, et al. (2010))

Our reply: We have revised Fig. 2a (see Fig. R1a) to help the reader envision how the scaffolds and bioreactor are positioned inside the mechanical stimulation unit (MSU). In the manuscript we added on p. 7:

‘Each bioreactor contained two scaffolds (Fig. 2a) and was mounted into a self-made mechanical stimulation unit (MSU)⁴⁰. The MSU allowed controlling the force F_{thres} to contact the scaffold, as well as frequency and strain (Fig. 2a) of the loading regime.’

23. *The media flow rate(s) and the frequency of media changes should be specified.*

Our reply: There was no media flow induced in the bioreactor. Medium change is now specified in the methods part p. 23: 'During the cell experiment, the cell culture medium was exchanged 3 times per week.'

REVIEWERS' COMMENTS:

Reviewer #1 (Remarks to the Author):

This manuscript has be improved significantly.

Reviewer #2 (Remarks to the Author):

The authors have satisfactorily addressed the reviewers' concerns and made major changes to the manuscript including new data that strengthens their findings. There are no further concerns about the manuscript.

Reviewer #3 (Remarks to the Author):

The authors prepared a thorough and comprehensive response to the previous review that addressed all of the critiques.